# PROVABLY EFFICIENT HIGH-ORDER FLOW MATCHING IN PIXEL SPACE

## ABSTRACT

We introduce high-order PixelFlow (HopeFlow), which is the first cascade flow model that learns both pixel-space velocity and acceleration fields end-to-end, lifting image generation beyond the limitations of purely first-order supervision. By incorporating second-order dynamics, HopeFlow aligns mid-horizon dependencies and high-curvature regions, yielding markedly smoother, more stable transport trajectories. The model trains directly on raw pixels—no VAE encoder-decoder is required—and remains computationally affordable. We prove that the HopeFlow model is computable by a $\mathsf{TC}^0$ class of threshold circuits, which operate with constant depth $O(1)$ and a polynomial number of gates $\mathrm{poly}(n)$. Moreover, by replacing exact attention with approximate attention layers, the end-to-end HopeFlow inference runs in almost quadratic time.

## 1 INTRODUCTION

Generative models (Esser et al., 2021; 2024; Labs, 2024; Podell et al., 2024; Ramesh et al., 2021; Rombach et al., 2022; Sun et al., 2024; Yang et al., 2025) have fundamentally reshaped visual content creation, driving innovation in creative design, media production, and digital content generation. Among the various approaches, diffusion-based models (Esser et al., 2024; Pernias et al., 2024; Podell et al., 2024; Rombach et al., 2022) have risen to prominence for their ability to synthesize high-quality images, videos, and 3D assets with remarkable realism and diversity. In particular, latent diffusion models (LDMs) (Ma et al., 2024; Peebles & Xie, 2023; Rombach et al., 2022), driven by the success of the Stable Diffusion (Rombach et al., 2022), have become the standard across multiple modalities. LDMs reduce computational overhead by encoding raw data into a compact latent space via pre-trained Variational Autoencoders (VAEs) (Kingma et al., 2013), facilitating efficient denoising throughout the diffusion process. Despite their widespread success, LDMs commonly treat the VAE and diffusion modules as distinct, decoupled components. This decoupling limits the potential for joint optimization and poses challenges for end-to-end interpretability and performance tuning.

PixelFlow (Chen et al., 2025b) departs from the latent-space paradigm by operating directly in raw pixel space, eliminating the need for pre-trained VAEs and separate upsamplers. It uses cascade flow-matching (Lipman et al., 2023; Liu et al., 2023b) across multiple resolutions in a single end-to-end model, dramatically reducing inference cost while preserving fidelity.

The limitation of PixelFlow is that its training objective matches only the instantaneous velocity (first derivative) along each flow trajectory. By supervising solely on

$$V_t = \frac{\mathrm{d}\mathsf{X}_t}{\mathrm{d}t}, \tag{1}$$

it overlooks higher-order dynamics, causing erratic trajectories and unstable transitions in regions of high curvature. To capture mid-horizon geometry, we extend the flow-matching loss to include the second derivative (acceleration)

$$A_t = \frac{\mathrm{d}^2\mathsf{X}_t}{\mathrm{d}t^2}, \tag{2}$$

enforcing alignment of both velocity and acceleration between model and data trajectories. This second-order supervision provides explicit curvature guidance, yielding smoother trajectories and more stable generative paths in complex regions.

Additionally, as image generation architectures become increasingly complex to meet high resolution, photorealistic demands, rigorous circuit-complexity analyses remain scarce. (Merrill & Sabharwal, 2023) show that DLOGTIME-uniform $TC^0$ circuits can simulate softmax-attention transformers; (Chen et al., 2024) prove RoPE adds no extra power; and (Ke et al., 2025) characterize the complexity of VAR. Yet whether pixel-space flow-matching augments VAR's expressive power is still open. Likewise, PixelFlow streamlines generation by eliminating separate VAEs and upsamplers and reducing inference cost, but lacks runtime analysis; we identify the attention mechanism as the primary bottleneck and accelerate it via low-rank approximation. We prove that our high-order pixel flow modules admit DLOGTIME-uniform $TC^0$ implementations. We also establish the worst case optimality for the HopeFlow. Lastly. we propose fast HopeFlow where its running time can be reduced to $O(n^{2+o(1)})$.

**Roadmap.** The remainder of the paper is organized as follows. Section 2 reviews related work on HopeFlow. Section 3 introduces fundamental concepts from circuit complexity that underpin our later analysis. Section 4 details the mathematical formalizations for all HopeFlow modules. Section 5 shows the circuit complexity and expressivity of HopeFlow. Section 6 shows that theoretical convergence rate of HopeFlow. Section 7 introduces a potential improvement to the HopeFlow architecture by replacing the attention layer with approximate attention layer. We conclude our paper in Section 8.

## 2 RELATED WORK

We survey prior work in three main areas: flow- and diffusion-based image generation in Section 2.1), theoretical circuit-complexity analyses of learning architectures in Section 2.2, and low-rank approximation techniques for accelerating transformer computations in Section 2.3.

### 2.1 FLOW-BASED AND DIFFUSION-BASED MODELS

VAEs have become a fundamental component in recent diffusion-based (Rombach et al., 2022; Podell et al., 2024; Yang et al., 2025) and flow-based models (Ren et al., 2025; Esser et al., 2024). To reduce computational costs, prior models commonly encode visual data into a lower-dimensional latent space using VAEs. However, this compression often sacrifices high-frequency details and information loss. This limitation sometime causes noticeable low-level artifacts in the generated results (Podell et al., 2024). Motivated by the goal of algorithmic simplicity and seamless end-to-end optimization, our model avoids reliance on VAEs and instead operates directly in the original pixel space.

Single-stage diffusion models (Balaji et al., 2022; Ho & Salimans, 2022; Sohl-Dickstein et al., 2015) work directly in pixel space and try to learn the full image distribution at once. While this works for low-resolution images, it becomes too slow and costly for high-resolution ones. To solve this, cascaded models (Gu et al., 2023; Ho et al., 2022; Kim et al., 2024; Saharia et al., 2022) were introduced. These models first create a low-resolution image and then use super-resolution steps to increase the quality. However, these steps often start from random noise and depend on the earlier output, making the process slow and hard to control. Also, since each stage is trained separately, the whole model cannot be optimized end-to-end and needs special tricks to make all parts work well together.

Recent research has revisited direct pixel-space generation with novel architectural innovations. Simple Diffusion (Hoogeboom et al., 2023; 2025) proposes a streamlined diffusion framework that achieves high performance on popular datasets such as ImageNet through refinements in model architecture and noise schedules. TarFlow (Zhai et al., 2025) introduces a Transformer-based normalizing flow architecture that is capable of generating image directly in the pixel space. The fractal generative models in FractalGen (Li et al., 2025a) achieves high quality pixel-wise generation by adopting recursive atomic module. PixelFlow (Chen et al., 2025b) achieves efficient pixel-space generation by using cascade flow matching across scales in a single model. Inspired by the potential of second-order dynamics, we propose HopeFlow, a model that learns pixel-space velocity and acceleration fields end-to-end, thereby lifting image generation beyond the limitations of purely first-order supervision and yielding markedly smoother, more stable transport trajectories.

Recent work has renewed interest in direct pixel-space generation by introducing new model designs. Simple Diffusion (Hoogeboom et al., 2023; 2025) simplifies the diffusion process and improves results on ImageNet through changes in the architecture and noise schedule. FractalGen (Li et al., 2025a) uses recursive building blocks to model images at the pixel level. TarFlow (Zhai et al., 2025) applies a Transformer-based normalizing flow to directly generate images in pixel space. PixelFlow (Chen et al., 2025b) improves efficiency by using flow matching across multiple scales in a single model. Building on this progress, we introduce HopeFlow, which learns both velocity and acceleration fields in pixel space. By including second-order dynamics and training the model end-to-end, HopeFlow goes beyond first-order methods and produces smoother and more stable image generation paths.

## 2.2 CIRCUIT COMPLEXITY

Circuit complexity studies the power of Boolean circuits by depth, size, and gate type, yielding the hierarchy $AC^0 \subset TC^0 \subseteq NC^1$, while the equality $TC^0 = NC^1$ remains open (Vollmer, 1999; Arora & Barak, 2009). This framework bounds machine-learning expressivity: non-uniform $TC^0$ can simulate AHATs (Merrill et al., 2022), L-uniform $TC^0$ can simulate SMATs (Liu et al., 2023a), and both admit DLOGTIME-uniform $TC^0$ approximations (Merrill & Sabharwal, 2023). Circuit-complexity techniques have been extended beyond standard Transformers to analyze a variety of other models (Chen et al., 2025c; Ke et al., 2025). These methods have been applied to alternative architectures, including state-space models and recurrent frameworks.

## 2.3 ACCELERATION VIA LOW RANK APPROXIMATION

Low-rank approximation has emerged as a powerful technique for addressing the computational challenges associated with modern transformer architectures. By approximating key operations such as attention and gradient computations, these methods significantly reduce the time and resource requirements of training and inference.

**Accelerating Attention Mechanisms.** Attention's quadratic cost in context length hinders scalability in modern LLMs (OpenAI, 2024; AI, 2024; Anthropic, 2024). Polynomial kernel approximations use low-rank factorizations to efficiently approximate the attention matrix (Aggarwal & Alman, 2022), bringing the per-layer computation for training and inference close to linear time (Alman & Song, 2023; 2024c). This approach has been extended to tensor attention while maintaining near-linear scaling (Alman & Song, 2024a), and adapted for differentially private cross-attention (Liang et al., 2024c) and RoPE-based attention (Alman & Song, 2024b). Complementary methods, such as the conv-basis transform (Liang et al., 2024a) and a range of pruning strategies (Li et al., 2024; Shen et al., 2025b;a) , further accelerate attention computation.

**Approximating the Gradient.** Low-rank approximation is a standard technique for reducing the computational cost of transformer training (Liang et al., 2024b;d; Alman & Song, 2024c; Hu et al., 2024; Chen et al., 2025a; Liang et al., 2024b; Li et al., 2025b). In particular, the forward-attention low-rank framework of (Alman & Song, 2023) is extended in (Alman & Song, 2024c) to approximate attention gradients, markedly lowering gradient-computation overhead. This gradient-approximation approach is then applied to multi-layer transformers in (Liang et al., 2024b), showing that backward passes can be executed in near-linear time. Meanwhile, (Liang et al., 2024d) extends the method in (Alman & Song, 2024c) to a tensor-based attention model by leveraging the forward-pass results of (Alman & Song, 2024a), speeding up the training time of tensorized attention. Lastly, the low-rank approximation is adopted in the training of Diffusion Transformers (Hu et al., 2024).

## 3 PRELIMINARY

We begin by fixing our core conventions in the Section 3.1. In Section 3.2, we then recall the DLOGTIME-uniform Boolean circuit hierarchies $NC^i$, $AC^i$, and $TC^i$ along with their uniformity requirements. Finally, we summarize known uniform $TC^0$ constructions for floating point operations in Section 3.3.

## 3.1 NOTATIONS

Let $X \in \mathbb{R}^{hw \times d}$ be a matrix, and denote its reshaped tensor form as $\mathsf{X} \in \mathbb{R}^{h \times w \times d}$. For any positive integer $n$, we use $[n]$ to denote the set $\{1, 2, \ldots, n\}$, and define the set of natural numbers as $\mathbb{N} := \{0, 1, 2, \ldots\}$. Consider a matrix $X \in \mathbb{R}^{m \times n}$, where $X_{i,j}$ denotes the element in the $i$-th row and $j$-th column. When $x_i \in \{0, 1\}^*$, it represents a binary string of arbitrary length. More generally, we use $x_i \in \{0, 1\}^p$ to denote a binary string of fixed length $p$, where each bit is either 0 or 1. For a matrix $X \in \mathbb{R}^{n \times d}$, we define its infinity norm as $\|X\|_\infty := \max_{i,j} |X_{i,j}|$, which corresponds to the maximum absolute value among all entries of $X$.

## 3.2 CIRCUIT COMPLEXITY CLASS

Boolean circuit is a directed acyclic graph of logic gates and is used to compute Boolean function. A language is a set of binary strings representing decision problems. A circuit family is L-uniform if its circuit descriptions can be generated by a log-space Turing machine, whereas DLOGTIME-uniformity requires a deterministic $O(\log n)$-time algorithm to decide each gate's type and wiring. The class $\mathsf{NC}^i$ comprises languages decidable by L-uniform families of polynomial-size and has depth $O((\log n)^i)$ circuits with bounded-fan-in gates. $\mathsf{NC}^i$ may consist bounded fan-in $\mathsf{AND}, \mathsf{OR}$ and unit fan-in $\mathsf{NOT}$. $\mathsf{AC}^i$ is defined similarly but allows unbounded-fan-in $\mathsf{AND/OR}$ gates; and $\mathsf{TC}^i$ further extends $\mathsf{AC}^i$ by including $\mathsf{MAJORITY}$ gate. We direct reader to Appendix A.1 for formal definitions.

## 3.3 CIRCUIT COMPLEXITY OF FLOATING-POINT ARITHMETIC

In this section, we will introduce circuit-complexity of standard floating-point operations. From (Chiang, 2025), basic arithmetic primitives and their iterated forms admit uniform $\mathsf{TC}^0$. We also show that one can approximate both the exponential and square-root functions to within relative error in uniform $\mathsf{TC}^0$. We denote by $d_{\mathrm{std}}, d_\otimes, d_\oplus, d_{\exp}$, and $d_{\mathrm{sqrt}}$ the depths required for standard arithmetic, iterated multiplication, iterated addition, exponential approximation, and square-root approximation, respectively. We direct reader to Appendix A.2 for formal definitions and proofs.

## 4 THE HOPEFLOW ARCHITECTURE

In this section, we give a precise mathematical formulation of the HopeFlow architecture. In Section 4.1 we introduce the core flow definitions and derive the velocity and acceleration fields at each scale. Section 4.2 then describes the training procedure, and Section 4.3 presents the inference algorithm. For brevity, the full derivations and module-by-module formulas are collected in Appendix B.

## 4.1 HOPEFLOW

We first introduce the core ideas of velocity field and acceleration field in the HopeFlow architecture.

**Definition 4.1** (HopeFlow). *Given the following:*

- **Input tensor:** $\mathsf{X} \in \mathbb{R}^{h \times w \times c}$ *where $h, w, c$ denote the height, width, and the number of channels, respectively.*

- **Number of scales:** $S \in \mathbb{N}$.

- **Downsampling function:** $\phi_{\mathrm{down}}(\cdot) : \mathbb{R}^{h \times w \times c} \to \mathbb{R}^{(h/r) \times (w/r) \times c}$ *from Defintion B.2.*

- **Upsampling function:** $\phi_{\mathrm{up}}(\cdot) : \mathbb{R}^{h \times w \times c} \to \mathbb{R}^{rh \times rw \times c}$ *from Definition B.1.*

- **Interpolation weights:** *functions $\alpha, \beta : [0, 1] \to \mathbb{R}$, with the following property:*

$$\alpha(t) + \beta(t) = 1, \alpha(0) = 0, \ \beta(0) = 1, \alpha(1) = 1, \ \beta(1) = 0,$$

*and $\alpha, \beta$ are continuously differentiable so that $\alpha', \beta', \alpha'', \beta''$ exist on $[0, 1]$.*

*Then, the model does the following:*

- **Stage times:** *For each scale $i \in [S]$, calculate $t_i^0 = (i-1)/S$ and $t_i^1 = i/S$.*

- **Noise tensor:** *For scale $i$, $\epsilon_i \in \mathbb{R}^{(h/2^i) \times (w/2^i) \times c}$ with every entry sampled from $\mathcal{N}(0, I)$.*

- **Coarse start state:** *For timestep $t \in [t_i^0, t_i^1]$, $\mathsf{F}_i^0 = t_i^0 \phi_{\text{up}}(\phi_{\text{down}}(\mathsf{X}, 2^{i+1}), 2) + (1 - t_i^0)\epsilon_i$ defining the start state of the flow from $t_i^0$ to $t_i^1$.*

- **Coarse end state:** *For timestep $t \in [t_i^0, t_i^1]$, $\mathsf{F}_i^1 = t_i^1 \phi_{\text{down}}(\mathsf{X}, 2^i) + (1 - t_i^1)\epsilon_i$ defining the end state of the flow from $t_i^0$ to $t_i^1$.*

- **Interpolation:** *For timestep $t \in [t_i^0, t_i^1]$ $F_i^t = \alpha(t)\mathsf{F}_i^1 + \beta(t)\mathsf{F}_i^0$. defining a trajectory between start state $\mathsf{F}_i^0$ to end state $\mathsf{F}_i^1$.*

- **Velocity field:** *The first-derivative of the flow at scale $i$ is $V_i^t = \frac{\mathrm{d}}{\mathrm{d}t}F_i^t = \alpha'(t)\,F_i^1 + \beta'(t)\,F_i^0$.*

- **Acceleration field:** *The second-derivative of the flow at scale $i$ is $A_i^t = \frac{\mathrm{d}^2}{\mathrm{d}t^2}F_i^t = \alpha''(t)\,F_i^1 + \beta''(t)\,F_i^0$.*

**Remark 4.2.** *The way PixelFlow and HopeFlow construct $F_i^0$ and $F_i^1$ is different from previous FlowAR. The major reason is we do not send the pixels to latent space.*

**Remark 4.3.** *Following previous work (Liu et al., 2023b), we set $\alpha(t) = e^{-\frac{1}{4}a(1-t)^2 - \frac{1}{2}b(1-t)}$ and $\beta(t) = \sqrt{1 - \alpha(t)^2}$, so that at each $t$, $F_i^t = \alpha(t)\,F_i^1 + \beta(t)\,F_i^0$. This choice ensures constant-variance interpolation—since $F_i^1$ and $F_i^0$ are independent, $\mathrm{Var}(F_i^t) = \alpha(t)^2\mathrm{Var}(F_i^1) + \beta(t)^2\mathrm{Var}(F_i^0)$, and $\alpha(t)^2 + \beta(t)^2 = 1$ keeps $\mathrm{Var}(F_i^t)$ fixed for all $t$. It also guarantees uniformly bounded derivatives: the explicit exponential form yields finite $\alpha'$, $\alpha''$, $\beta'$, and $\beta''$ on $[0, 1]$, avoiding the endpoint singularities or unbounded curvature that would arise under a polynomial or piecewise-linear schedule.*

We define the first-order HopeFlow architecture (FlowF) and second-order architecture (FlowH). For brevity, we direct reader to Appendix B.3.

**Definition 4.4** (First-Order HopeFlow Architecture, informal version of Definition B.8)**.** *Let $X \in \mathbb{R}^{h \times w \times c}$ be the input tensor (height $h$, width $w$, channels $c$), $S \in \mathcal{N}$ the number of scales with base factor $a \in \mathcal{N}^+$ and scale factors $r_i = a^{K-i}$. For each $i \in [S]$, let $\mathsf{F}_i^1 \in \mathbb{R}^{(h/r_i) \times (w/r_i) \times c}$ be the downsampled end state, $\mathsf{F}_i^t$ the interpolated state at time $t_i \in [(i-1)/S, i/S]$, and let $\mathsf{Attn}_i$, $\mathsf{MLP}_i(\cdot, c, d)$, $\mathsf{LN}_i$ denote the attention, MLP, and layer-norm layers. The layer $\mathsf{FFlowH}_i$ computes:*

$$(\alpha_1, \alpha_2, \beta_1, \beta_2, \gamma_1, \gamma_2) = \mathsf{MLP}_i(\mathsf{F}_i^1 + t_i\mathbf{1}, c, 6c),$$
$$\mathsf{F}_i'^t = \mathsf{Attn}_i(\gamma_1 \circ \mathsf{LN}_i(\mathsf{F}_i^t) + \beta_1) \circ \alpha_1,$$
$$\mathsf{F}_i''^t = \mathsf{MLP}_i(\gamma_2 \circ \mathsf{LN}_i(\mathsf{F}_i'^t) + \beta_2, c, c) \circ \alpha_2,$$

*so that $\mathsf{F}_i''^t = \mathsf{FlowF}_i(\mathsf{F}_i^1, \mathsf{F}_i^t, t_i)$.*

**Definition 4.5** (Second-Order HopeFlow Architecture, informal version of Definition B.9)**.** *Let $X \in \mathbb{R}^{h \times w \times c}$ be the input tensor (height $h$, width $w$, channels $c$), $S \in \mathcal{N}$ the number of scales with base factor $a \in \mathcal{N}^+$ and scale factors $r_i = a^{K-i}$. For each $i \in [S]$, let $\mathsf{F}_i^1 \in \mathbb{R}^{(h/r_i) \times (w/r_i) \times c}$ be the downsampled end state, $\mathsf{F}_i^t$ the interpolated state at time $t_i \in [(i-1)/S, i/S]$, $\mathsf{F}_i^{\text{first}}$ the first order HopeFlow, and let $\mathsf{Attn}_i$, $\mathsf{MLP}_i(\cdot, c, d)$, $\mathsf{LN}_i$ denote the attention, MLP, and layer-norm layers. The layer $\mathsf{FFlowH}_i$ computes:*

$$(\alpha_1, \alpha_2, \beta_1, \beta_2, \gamma_1, \gamma_2) = \mathsf{MLP}_i(\mathsf{F}_i^1 + t_i\mathbf{1}, c, 6c),$$
$$\mathsf{F}_i'^t = \mathsf{MLP}_i(\mathsf{Concat}(\mathsf{F}_i^t, \mathsf{F}_i^{\text{first}}), 2, 1),$$
$$\mathsf{F}_i''^t = \mathsf{Attn}_i(\gamma_1 \circ \mathsf{LN}_i(\mathsf{F}_i'^t) + \beta_1) \circ \alpha_1,$$
$$\mathsf{F}_i'''^t = \mathsf{MLP}_i(\gamma_2 \circ \mathsf{LN}_i(\mathsf{F}_i''^t) + \beta_2, c, c) \circ \alpha_2,$$

*so that $\mathsf{F}_i'''^t = \mathsf{FlowH}_i(\mathsf{F}_i^1, \mathsf{F}_i^t, \mathsf{F}_i^{\text{first}}, t_i)$.*

## 4.2 TRAINING OF HOPEFLOW ARCHITECTURE

This subsection details the training procedure of the HopeFlow architecture, which operates across multiple spatial scales to learn both first-order and second-order flow representations. At each scale, the model is trained to reconstruct intermediate representations from noisy inputs using ground-truth signals derived from the image data. The full training routine is summarized in Algorithm 1.

---

**Algorithm 1** HopeFlow Training

---

1: **procedure** HOPEFLOWTRAINING($\theta, D, S, \{t_s^0, t_s^1\}_{s=1}^S$)
2:     /* $\theta$ denotes the model parameters of FlowF, FlowH */
3:     /* $D$ denotes the training dataset. */
4:     /* $S$ denotes the number of scale */
5:     /* $\{t_s^0, t_s^1\}$ denotes the start/end times for each scale */
6:     **while** not converge **do**
7:        $\mathsf{X}_{\text{img}} \sim D$             ▷ Sample an image from dataset.
8:        $\ell \leftarrow 0$              ▷ Init loss as 0.
9:        **for** $s = 0 \rightarrow (S - 1)$ **do**          ▷ Train the model on $S$ scales.
10:           $\epsilon \sim \mathcal{N}(0, I)$          ▷ Sample random noise.
11:           $t \sim [0, 1]$          ▷ Sample a random timestep.
12:           $\mathsf{F}^0 \leftarrow t_s^0 \phi_{\text{up}}(\phi_{\text{down}}(\mathsf{X}_{\text{img}}, 2^{s+1}), 2) + (1 - t_s^0)\epsilon$    ▷ Calculate start state.
13:           $\mathsf{F}^1 \leftarrow t_s^1 \phi_{\text{down}}(\mathsf{X}_{\text{img}}, 2^s) + (1 - t_s^1)\epsilon$      ▷ Calculate end state.
14:           $\mathsf{F}_{\text{noise}}^t \leftarrow \alpha_t \mathsf{F}^1 + \beta_t \mathsf{F}^0$       ▷ Calculate noisy input.
15:           $\mathsf{F}_{\text{first}}^t \leftarrow \alpha_t' \mathsf{F}^1 + \beta_t' \mathsf{F}^0$     ▷ Calculate first-order ground-truth.
16:           $\mathsf{F}_{\text{second}}^t \leftarrow \alpha_t'' \mathsf{F}^1 + \beta_t'' \mathsf{F}^0$     ▷ Calculate second-order ground-truth.
17:           $\widehat{\mathsf{F}}_{\text{first}}^t \leftarrow \mathsf{FlowF}(\mathsf{F}_{\text{noise}}^t, \mathsf{F}^1)$         ▷ Predict FlowF.
18:           $\widehat{\mathsf{F}}_{\text{second}}^t \leftarrow \mathsf{FlowH}(\mathsf{F}_{\text{noise}}^t, \mathsf{F}^1, \widehat{\mathsf{F}}_{\text{first}}^t)$      ▷ Predict FlowH.
19:           $\ell_c \leftarrow \|\widehat{\mathsf{F}}_{\text{first}}^t - \mathsf{F}_{\text{first}}^t\|_2^2 + \|\widehat{\mathsf{F}}_{\text{second}}^t - \mathsf{F}_{\text{second}}^t\|_2^2$     ▷ Calculate loss.
20:           $\ell \leftarrow \ell + \ell_c$
21:        **end for**
22:        $\theta \leftarrow \nabla_\theta \ell$           ▷ Optimize parameter $\theta$ with $l$.
23:     **end while**
24:     **return** $\theta$
25: **end procedure**

---

## 4.3 INFERENCE OF HOPEFLOW ARCHITECTURE

This subsection describes the inference process of HopeFlow architecture. Starting from pure Gaussian noise, the model iteratively refines the image across multiple scales and timesteps by integrating both first- and second-order flow predictions. These predictions are generated by learned flow matching modules and applied to progressively denoise and reconstruct the image. The complete inference routine is outlined in Algorithm 2.

## 5 COMPLEXITY OF HOPEFLOW ARCHITECTURE

In this section, we show the circuit-complexity bounds for HopeFlow. This relies on key results about the complexity of their fundamental modules. For brevity, we refer the reader to Appendix C for necessary theorems and proofs. Here, we present the main result.

We first prove that the first-order HopeFlow layer, FlowF, defined in Definition B.8 can be efficiently simulated by a uniform $\mathsf{TC}^0$ circuit.

**Lemma 5.1** (First-order HopeFlow layer computation in $\mathsf{TC}^0$, informal version of Lemma C.8)**.** $\mathsf{FlowF}(\mathsf{X})$*, as defined in Definition B.8, lies in the uniform* $\mathsf{TC}^0$ *class with depth* $26d_{\text{std}} + 12d_{\oplus} + 2d_{\text{sqrt}} + d_{\exp}$ *and size* $\text{poly}(n)$.

---

**Algorithm 2** HopeFlow Inference

---

1: **procedure** HOPEFLOWINFERENCE($\theta, S, T$)
2:     /* $\theta$ denotes the model parameters of FlowF, FlowH */
3:     /* $S$ denotes the number of scale */
4:     /* $T$ denotes the number of timesteps for each scale */
5:     $X_{img} \leftarrow \mathcal{N}(0, I)$                                   ▷ Init the $X_{img}$ with random noise.
6:     $\Delta t \leftarrow 1/T$                                        ▷ Calculate the step size $\Delta t$
7:     **for** $s = 1 \rightarrow S$ **do**                            ▷ Inference through $S$ stages.
8:         **for** $t = 1 \rightarrow T$ **do**
9:             $\widehat{F}_{first} \leftarrow FlowF(X_{img})$               ▷ Calculate FlowF output.
10:             $\widehat{F}_{second} \leftarrow FlowH(X_{img}, \widehat{F}_{first})$      ▷ Calculate FlowH output.
11:             $X_{img} \leftarrow X_{img} + \widehat{F}_{first} \cdot \Delta t + 0.5 \cdot \widehat{F}_{second} \cdot (\Delta t)^2$     ▷ Apply terms.
12:         **end for**
13:         **if** $s \neq S$ **then**
14:             $X_{img} \leftarrow \phi_{up}(X_{img}, 2)$                   ▷ Upsample $X_{img}$.
15:         **end if**
16:     **end for**
17:     **return** $X_{img}$                               ▷ Return the final image.
18: **end procedure**

---

The $d_{std}$, $d_{\oplus}$, $d_{exp}$, and $d_{sqrt}$ are defined in Definition A.9, Definition A.10, and Definition A.11 respectively. Then, we prove that the second-order HopeFlow layer, FlowH, defined in Definition B.9 can be simulated by a uniform $\mathsf{TC}^0$ circuit.

**Lemma 5.2** (Second-order HopeFlow layer computation in $\mathsf{TC}^0$, informal version of Lemma C.9)**.** FlowH(X)*, as defined in Definition B.9, lies in the uniform* $\mathsf{TC}^0$ *class with depth* $28d_{std} + 13d_{\oplus} + 2d_{sqrt} + d_{exp}$ *and size* $\mathrm{poly}(n)$.

With Lemma 5.1 and Lemma 5.2, we show the HopeFlow models can be simulated by a uniform $\mathsf{TC}^0$ circuit.

**Theorem 5.3** (HopeFlow computation in $\mathsf{TC}^0$)**.** *Suppose the precision* $p \in O(\mathrm{poly}(n))$, $X \in \mathbb{R}^{h \times w \times c}$, $n = h = w$, $r \leq n$, $c \leq n$, *and* $K = O(1)$. *The* $d_{std}$, $d_{\oplus}$, $d_{exp}$, *and* $d_{sqrt}$ *are defined in Definition A.9, Definition A.10, and Definition A.11 respectively. Then the HopeFlow model lies in the uniform* $\mathsf{TC}^0$ *circuit family.*

*Proof.* By Lemma C.2, Lemma C.3, Lemma C.6, Lemma C.5, C.8 and Lemma C.9, each layer in HopeFlow Model lies in the uniform $\mathsf{TC}^0$ circuit with size $\mathrm{poly}(n)$ and depth $O(1)$. Since there exist finite $K = O(1)$ layers, the composition of $K$ circuit also lies in the uniform $\mathsf{TC}^0$ circuit with size $\mathrm{poly}(n)$ and depth $O(1)$. $\square$

Theorem 5.3 shows that a DLOGTIME-uniform $\mathsf{TC}^0$ circuit family can simualte a HopeFlow model with $\mathrm{poly}(n)$ precision, constant depth, and $\mathrm{poly}(n)$ size. Inference therefore runs in $O(1)$ parallel time using only polynomially many simple threshold gates, and the wiring for each input size can be generated in $O(\log n)$ time. As a result, even with its high-order components, HopeFlow remains maximally parallelizable, low-latency, and hardware-friendly.

## 6   STATISTICAL CONVERGENCE GUARANTEES OF HOPEFLOW

In this section, we show that the HopeFlow architecture inherits worst-case optimal convergence rates for learning both velocity and acceleration fields. Our main theorem is taken from (Gong et al., 2025) and depends on a handful of key assumptions (Assumptions D.7, D.9, D.10, D.11, D.12, D.13, D.14). Below, we verify that HopeFlow satisfies each of these assumptions.

We assume the true data distribution $p_0$ is supported on $[-1, 1]^d$. In practice, we linearly rescale pixel-values from $[0, 255]^d$ into $[-1, 1]^d$. Moreover, we assume $p_0 \in B_{p,q}^s([-1, 1]^d)$ for some

$s > 0, p \geq 1, q \geq 1$. Equivalently, $p_0$ has $s$-order Besov regularity; any compactly supported $C^s$ density satisfies this. Thus Assumption D.7 holds.

**Remark 6.1.** *Convolution with a nondegenerate Gaussian of width $\sigma > 0$ guarantees that the "blurred" image density is infinitely differentiable. Consequently, $p_0$ (after optional Gaussian blur) lies in $B_{p,q}^s$ for every $s > 0$ and $1 \leq p, q \leq \infty$. Thus, assuming $p_0 \in B_{p,q}^s$ is no stronger than assuming each true image density is $C^s$.*

We provide a summary of the assumptions in (Gong et al., 2025) on $\alpha(t), \beta(t)$ pairs. The interpolation weights $\alpha(t), \beta(t)$ are required to satisfy the following: as $t \to 0$, one has $\alpha_t = b_0, t^\kappa$ and $1 - \beta_t = \widetilde{b}_0, t^{\bar{\kappa}}$ for some $\kappa, \bar{\kappa} > 0$, ensuring $\alpha_0 = 0$ and $\beta_0 = 1$; for all $t \in [T_0, 1]$, there is a constant $D_0 \geq 1$ such that $D_0^{-1} \leq \alpha_t^2 + \beta_t^2 \leq D_0$; their first and second derivatives obey $|\alpha'(t)| + |\beta'(t)| \leq N, K_0$ and $|\alpha''(t)| + |\beta''(t)| \leq N, K_0$ for some $K_0 > 0$; and, when $\kappa = 1/2$, there exist $b_1, D_1 > 0$ so that for any $0 \leq \gamma < R_0$, one has $\int_{T_0}^{N^{-\gamma}} (\alpha'(t)^2 + \beta'(t)^2), dt \leq D_1 \log^{b_1} N$ and $\int_{T_0}^{N^{-\gamma}} (\alpha''(t)^2 + \beta''(t)^2), dt \leq D_1 \log^{b_1} N$. If our $\alpha(t), \beta(t)$ satisfy these same conditions, then the Theorem 6.2 and Theorem 6.3 hold for HopeFlow.

**Theorem 6.2** (Bound Acceleration Error under Small $t$, Theorem 4.1 on page 9 in (Gong et al., 2025)). *If the following conditions hold: 1) Assume Assumption D.7, D.8, D.9, D.11, D.13, D.14 hold. 2) Let $C_6$ be a constant independent of $t$. 3) Let $x_1$ be the trajectory, $x_2 := \phi_1(x_1, t)$ where $\phi_1$ is the neural network in Lemma D.5. 4) Let $x$ be defined as the concatenation of $x_1$ and $x_2$, i.e., $x := [x_1, x_2]$.*

*Then there is a neural network $u_1 \in \mathcal{M}(L, W, S, B)$ and a constant $C$, which is independent of $t$, such that, for sufficiently large $N$,*

$$\int \|u_1(x, t) - a_t(x_1)\|_2^2 \cdot p_t(x_1) \mathrm{d}x_1 \leq C_6 \cdot (\alpha_t''^2 \log N + \beta_t''^2) \cdot N^{-\frac{2s}{d}},$$

*for any $t \in [T_0, 3T_*]$, where $L = O(\log^4 N), \|W\|_\infty = O(N \log^6 N), S = O(N \log^8 N), B = \exp(O((\log N) \cdot (\log \log N)))$.*

**Theorem 6.3** (Bound Acceleration Error under Large $t$, Theorem 4.2 on page 9 in (Gong et al., 2025)). *If the following conditions hold: 1) Fix $t_* \in [T_*, 1]$ and take arbitrary $\eta > 0$. 2) Assume Assumption D.7, D.8, D.9, D.11, D.13, D.14 hold. 3) Let $C_7$ be a constant independent of $t$. 4) Let $x_1$ be the trajectory, $x_2 := \phi_2(x_1, t)$ where $\phi_2$ is the neural network in Lemma D.6. 5) Let $x$ be defined as the concatenation of $x_1$ and $x_2$, i.e., $x := [x_1, x_2]$.*

*Then there is a neural network $u_2 \in \mathcal{M}(L, W, S, B)$ and a constant $C$, which is independent of $t$, such that, for sufficiently large $N$,*

$$\int \|u_2(x, t) - a_t(x_1)\|_2^2 \cdot p_t(x_1) \mathrm{d}x_1 \leq C_7 \cdot ((\alpha_t'')^2 \log N + (\beta_t'')^2) \cdot N^{-\eta},$$

*for any $t \in [2t_*, 1]$, where $L = O(\log^4 N), \|W\|_\infty = O(N \log^6 N), S = O(N \log^8 N), B = \exp(O((\log N) \cdot (\log \log N)))$.*

**Remark 6.4.** *In particular, $\alpha'$, $\alpha''$, $\beta'$, and $\beta''$ are all finite for every $t \in [0, 1]$, so we avoid the endpoint singularities or unbounded curvature that would arise under a polynomial or piecewise-linear schedule.*

## 7 FAST HOPEFLOW

In this section, we propose a potential improvement to the HopeFlow architecture by replacing the attention layer with approximate attention layer. Section 7.1 introduces the approximate attention. In section 7.2, we show the Fast HopeFlow by replacing the original attention layers with approxmiate attention layers. Lastly, we provide a formal analysis on inference running time of HopeFlow and Fast HopeFlow in Section 7.3. We show that, with the approximate attention, we can reduce the inference runtime from $O(n^{4+o(1)})$ to $O(n^{2+o(1)})$.

### 7.1 APPROXIMATE ATTENTION COMPUTATION

To improve the efficiency of attention computation, we introduce an approximate method that guarantees a controlled error.

**Definition 7.1** (Approximate Attention Computation $\mathsf{AAttC}(n, d, R, \delta)$, Definition 1.2 in (Alman & Song, 2023))**.** *Let $X \in \mathbb{R}^{n \times d}$ represent the input sequence, and let $\delta > 0$ denote the allowed approximation error. Suppose $Q$, $K$, and $V$ are projection matrices in $\mathbb{R}^{n \times d}$, each with row norms bounded above by a constant $R$, i.e., $\max\{\|Q\|_\infty, \|K\|_\infty, \|V\|_\infty\} \leq R$, The procedure $\mathsf{AAttC}(n, d, R, \delta)$ returns an output $N \in \mathbb{R}^{n \times d}$ that approximates the true attention output $\mathsf{Attn}(X)$ with entrywise error bounded as $\|N - \mathsf{Attn}(X)\|_\infty \leq \delta$.*

We now give the runtime analysis of $\mathsf{AAttC}$.

**Lemma 7.2** (Subquadratic Runtime for Approximate Attention (Theorem 1.4 in (Alman & Song, 2023)))**.** *Consider the approximate attention mechanism $\mathsf{AAttC}$ defined in Definition 7.1. When the embedding dimension is set as $d = O(\log n)$, the bound on weight norms is $R = \Theta(\sqrt{\log n})$, and the approximation tolerance is $\delta = 1/\mathrm{poly}(n)$, then the time required to compute $\mathsf{AAttC}$ satisfies $\mathcal{T}(n, n^{o(1)}, d) = n^{1+o(1)}$, where $\mathcal{T}$ denotes the total runtime under the specified parameter settings.*

### 7.2 FAST HOPEFLOW ARCHITECTURE

Using the approximate attention, we define the fast first-order HopeFlow and second-order HopeFlow layer by replacing the original attention layers with $\mathsf{AAttC}$ layers. We give formal definition of fast first-order HopeFlow layer (FFlowF) in Definition E.1 and fast second-order HopeFlow layer (FFlowH) in Definition E.2 in Appendix E.1.

### 7.3 RUNNING TIME

In this section, we formally analyze the running time complexity of HopeFlow and Fast HopeFlow during inference. In Lemma 7.3, we show the total inference runtime of HopeFlow is bounded by $O(n^{4+o(1)})$.

**Lemma 7.3** (Inference Runtime of Original HopeFlow Architecture, informal version of Lemma E.3)**.** *Consider the original HopeFlow inference pipeline with the following parameters: Let $\mathsf{X} \in \mathbb{R}^{h \times w \times c}$ be the input tensor, and let $K = O(1)$ be the number of scales. For each scale $i \in [K]$ and a base factor $a \in \mathbb{N}^+$, the scale factor is defined as $r_i := a^{K-i}$. The architecture includes upsampling functions $\phi_{\mathrm{up}, i}(\cdot, a)$ (Definition B.1), attention layers $\mathsf{Attn}_i(\cdot)$ (Definition B.3), feed-forward layers $\mathsf{FFN}_i(\cdot)$ (Definition B.5), first-order HopeFlow layers $\mathsf{FlowF}_i(\cdot, \cdot, \cdot)$ (Definition B.8), and second-order HopeFlow layers $\mathsf{FlowH}_i(\cdot, \cdot, \cdot, \cdot)$ (Definition B.9). Then, HopeFlow achieves inference in $O(n^{4+o(1)})$ time.*

The total runtime is dominated by the running time of attention layers in HopeFlow. With approximate attention in Lemma 7.4, the total inference runtime of Fast HopeFlow is reduced to $O(n^{2+o(1)})$.

**Lemma 7.4** (Inference Runtime of Fast HopeFlow Architecture, informal version of Lemma E.4)**.** *Consider the Fast HopeFlow inference pipeline with the following parameters: Let $\mathsf{X} \in \mathbb{R}^{h \times w \times c}$ be the input tensor, and let $K = O(1)$ be the number of scales. For each scale $i \in [K]$ and a base factor $a \in \mathbb{N}^+$, the scale factor is defined as $r_i := a^{K-i}$. The architecture includes upsampling functions $\phi_{\mathrm{up}, i}(\cdot, a)$ (Definition B.1), approximate attention layers $\mathsf{AAttC}_i(\cdot)$ (Definition 7.1), feed-forward layers $\mathsf{FFN}_i(\cdot)$ (Definition B.5), first-order Fast HopeFlow layers $\mathsf{FFlowF}_i(\cdot, \cdot, \cdot)$ (Definition E.1), and second-order Fast HopeFlow layers $\mathsf{FFlowH}_i(\cdot, \cdot, \cdot, \cdot)$ (Definition E.2). Then, Fast HopeFlow achieves inference in $O(n^{2+o(1)})$ time.*

## 8 CONCLUSION

We have presented HopeFlow, a high-order flow-matching architecture that extends PixelFlow by incorporating multi-scale, high-order coupling to better capture complex dependencies in visual data. Our theoretical analysis shows that HopeFlow retains the $\mathsf{TC}^0$ circuit complexity and statistical convergence guarantees. Furthermore, we also propose a fast HopeFlow architecture. We hope HopeFlow will inspire more future research in high-order supervision image modeling.

ETHIC STATEMENT

This paper does not involve human subjects, personally identifiable data, or sensitive applications. We do not foresee direct ethical risks. We follow the ICLR Code of Ethics and affirm that all aspects of this research comply with the principles of fairness, transparency, and integrity.

REPRODUCIBILITY STATEMENT

We ensure reproducibility of our theoretical results by including all formal assumptions, definitions, and complete proofs in the appendix. The main text states each theorem clearly and refers to the detailed proofs. No external data or software is required.

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

# Appendix

**Roadmap.** In Section A, we present all notations and background of this paper. In Section B, we give a formal definition to the modules in HopeFlow. In Section C, we show the supplementary results on the circuit complexity of fundamental modules in the HopeFlow architecture. In Section D, we establish the statistical convergence guarantees for the HopeFlow architecture. In Section E, we provide supplementary definitions and time complexity proofs on fast HopeFlow architecture.

## A PRELIMINARY

This section introduces foundational concepts and notation used throughout the paper. We review standard Boolean circuit complexity classes, including $\mathsf{NC}^i$, $\mathsf{AC}^i$, and $\mathsf{TC}^i$, along with notions of uniformity such as L-uniformity and DLOGTIME-uniformity. Then, we define floating-point representations and present known results on computing basic arithmetic operations, exponentials, and square roots within constant-depth $\mathsf{TC}^0$ circuits.

### A.1 CIRCUIT COMPLEXITY CLASS

The following definitions are related to circuit complexity.

**Definition A.1** (Boolean Circuit, page 102 of (Arora & Barak, 2009)). *A Boolean circuit $C$ with $n$ inputs and $m$ outputs is a directed acyclic graph, for every $n, m \in \mathbb{N}$. The Boolean circuit satisfies the following structure:*

- *The $n$ input nodes have no incoming edges.*

- *The $m$ output nodes have no outgoing edges.*

- *All other nodes are called gate and labeled with one of* AND, OR, NOT.

- *All non-input nodes compute Boolean functions by applying their respective logical operations to the values received along their incoming edges.*

We next provide the formal definition of languages associated with a specific Boolean circuit.

**Definition A.2** (Languages Recognition, page 103 of (Arora & Barak, 2009)). *Given a Boolean circuit family $\mathcal{C} = \{C_n\}_{n \in \mathbb{N}}$, we say that a language $L \subseteq \{0, 1\}^*$ is recognized by the $\mathcal{C}$ if the following conditions hold:*

- *Length-specific circuits: Each circuit $C_n$ in the family takes $n$-bit inputs, i.e., $C_n : \{0, 1\}^n \to \{0, 1\}$.*

- *Correct recognition: For every string $x \in \{0, 1\}^*$, we have $x \in L$ if and only if $C_{|x|}(x) = 1$.*

- *Completeness: The family includes one circuit $C_n$ for every input length $n \in \mathbb{N}$.*

We now present various classes of languages that can be recognized by different families of Boolean circuits. We begin with the class $\mathsf{NC}^i$.

**Definition A.3** ($\mathsf{NC}^i$, page 110 of (Arora & Barak, 2009)). *A language is in $\mathsf{NC}^i$ if the following conditions hold:*

- *All gates have maximum fan-in of $2$.*

- *It can be decided by Boolean ciruit with size $O(\mathrm{poly}(n))$ and depth $O((\log n)^i)$*

Another class related to $\mathsf{NC}^i$ is $\mathsf{AC}^i$.

**Definition A.4** ($\mathsf{AC}^i$, page 110 of (Arora & Barak, 2009)). *A language is in $\mathsf{AC}^i$ if the following conditions hold:*

- *All gates are allowed to have unbounded fan-in.*

- *It can be decided by Boolean ciruit with size $O(\text{poly}(n))$ and depth $O((\log n)^i)$.*

$\mathsf{TC}^i$ extends $\mathsf{AC}^i$ by introducing the MAJORITY gate. This gate evaluates to false when half or more arguments are false and true otherwise.

**Definition A.5** ($\mathsf{TC}^i$, (Vollmer, 1999)). *A language is in $\mathsf{TC}^i$ if the following conditions hold:*

- *All gates are allowed to have unbounded fan-in.*

- *It can be decided by threshold ciruit, which are Boolean circuits with* AND, OR, MAJORITY, *with size $O(\text{poly}(n))$ and depth $O((\log n)^i)$*

In this paper, we refer to any Boolean circuit that includes MAJORITY gates as a *threshold circuit*. We now introduce two important notions of uniformity: L-uniformity and DLOGTIME-uniformity.

**Definition A.6** (L-uniform Circuit Families, page 104 of (Arora & Barak, 2009)). *For a circuit familar $\{C_n\}$, if there exist a deterministic log-space computable function that takes $1^n$ as input and outputs the description of the circuit $C_n$, then $\{C_n\}$ is L-uniform.*

We now present the definition of DLOGTIME-uniformity.

**Definition A.7** (DLOGTIME-uniform Circuit Families, (Barrington & Immerman, 1994)). *For a circuit familar $\{C_n\}$, if there exist a random-access log-space computable function that takes $1^n$ as input and outputs the description of the circuit $C_n$, then $\{C_n\}$ is DLOGTIME-uniform.*

### A.2 CIRCUIT COMPLEXITY OF FLOATING-POINT ARITHMETIC

In this section, we introduce the complexity of floating-point number operations.

**Definition A.8** ($p$-Bit Floating-Point Number, (Chiang, 2025)). *Given a pair $\langle a, b \rangle$ where $a$ is called significand and $b$ is called exponent, the $p$-bit floating-point number is defined as follows:*

- $a \in (-2^p, -2^{p-1}] \cup \{0\} \cup [2^{p-1}, 2^p).$

- $b \in [-2^p, 2^p).$

*This pair represents the real number $a \cdot w^b$, where $w$ is a fixed base. $\mathsf{F}_p$ represents the set of all $p$-bit floating-point numbers.*

The following definition shows the circuit complexity bounds for some floating-point operations.

**Lemma A.9** (Floating-point operations in $\mathsf{TC}^0$, (Chiang, 2025)). *Given floating-point numbers with $p \in O(\text{poly}(n))$, the addition, comparison, division, and multiplication between two numbers, product and sum between $n$ numbers can be computed by uniform $\mathsf{TC}^0$ circuits. We define the circuit depth of these operations as follows:*

- *The circuit depth of addition, comparison, division, and multiplication between two numbers is denoted as $d_{\text{std}}$.*

- *The circuit depth of sum between $n$ numbers is denoted as $d_\oplus$.*

- *The circuit depth of product between $n$ numbers is denoted as $d_\otimes$.*

**Lemma A.10** (Exponential approximation in $\mathsf{TC}^0$, (Chiang, 2025)). *Given floating-point number $x$ with $p \in O(\text{poly}(n))$, the $\exp(x)$ can be computed in $\mathsf{TC}^0$ with a relative error of at most $2^{-p}$. The circuit depth of $\exp(x)$ is denoted as $d_{\text{exp}}$.*

**Lemma A.11** (Square root approximation in $\mathsf{TC}^0$, (Chiang, 2025)). *Given floating-point number $x$ with $p \in O(\text{poly}(n))$, the $\sqrt{x}$ can be rounded to the nearest floating-point number and computed in $\mathsf{TC}^0$. The circuit depth of $\sqrt{x}$ is denoted as $d_{\text{sqrt}}$.*

## B THE HOPEFLOW ARCHITECTURE

This section provides a formal mathematical specification of the HopeFlow architecture. We begin by defining the fundamental sample operations, including nearest upsampling and bilinear downsampling. Next, we describe the building blocks of the autoregressive transformer, such as attention,

MLP, feedforward, and normalization layers, and then formally define the autoregressive transformer itself. We then introduce the full HopeFlow model, presenting both its first-order (Definition B.8) and second-order (Definition B.9) variants.

### B.1 SAMPLE FUNCTION

We start with the upsampling and downsampling functions.

**Definition B.1** (Nearest Upsampling Function). *The upsampling function $\phi_{\mathrm{up}}(\mathsf{X}, r)$ takes $\mathsf{X} \in \mathbb{R}^{h \times w \times c}$ and $r \geq 1$ as input and returns $\mathsf{Y} \in \mathbb{R}^{rh \times rw \times c}$. Specifically, for $i \in [rh], j \in [rw], l \in [c]$,*

$$\mathsf{Y}_{i,j,l} = \mathsf{X}_{\lfloor \frac{i}{r} \rfloor, \lfloor \frac{j}{r} \rfloor, l}.$$

Next, we define the bilinear downsampling function used in HopeFlow.

**Definition B.2** (Bilinear Downsampling Function). *The downsampling function $\phi_{\mathrm{down}}(\mathsf{X}, r)$ takes $\mathsf{X} \in \mathbb{R}^{h \times w \times c}$, $r \geq 1$, and a bilinear transformation matrix $\Phi_{down} \in \mathbb{R}^{(h/r \cdot w/r) \times hw}$ as input and returns $\mathsf{Y} \in \mathbb{R}^{(h/r) \times (w/r) \times c}$. Specifically, $\phi_{\mathrm{down}}(\mathsf{X}, r)$ does the following:*

- *Collapse the spatial dimensions of the input tensor $\mathsf{X}$ into a matrix $X \in \mathbb{R}^{hw \times c}$.*

- *Multiply by the downsampling operator to get*

$$Y = \Phi_{\mathrm{down}} X.$$

- *restore its spatial layout by reshaping $Y$ into $\mathsf{Y} \in \mathbb{R}^{(h/r) \times (w/r) \times c}$.*

### B.2 AUTOREGRESSIVE TRANSFORMER

In this section, we list formal definitions of each layer within the autoregressive transformer.

**Definition B.3** (Attention Layer). *The attention layer $\mathsf{Attn}(\mathsf{X})$ takes $\mathsf{X} \in \mathbb{R}^{h \times w \times c}$ and returns $\mathsf{Y} \in \mathbb{R}^{h \times w \times c}$. Specifically, $\mathsf{Attn}(\mathsf{X})$ does the following:*

- *Collapse the spatial dimensions of the input tensor $\mathsf{X}$ into a matrix $X \in \mathbb{R}^{hw \times c}$.*

- *Compute the unnormalized attention scores:*

$$A_{i,j} := \exp\left(X_{i,*} W_Q W_K^\top X_{j,*}^\top\right), \forall i, j \in [hw].$$

  *where $W_Q, W_K, W_V \in \mathbb{R}^{c \times c}$ are query, key, and value projection matrices.*

- *Compute output matrix:*

$$Y := D^{-1} A X W_V \in \mathbb{R}^{hw \times c}.$$

  *where $D = \mathrm{diag}(A \mathbf{1}_n) \in \mathbb{R}^{hw \times hw}$.*

- *Restore its spatial layout by reshaping $Y$ into $\mathsf{Y} \in \mathbb{R}^{h \times w \times c}$.*

We now define the multi-layer perceptron (MLP) layer.

**Definition B.4** (MLP Layer). *The MLP layer $\mathsf{MLP}(\mathsf{X}, c, d)$ takes $\mathsf{X} \in \mathbb{R}^{h \times w \times c}$ as input and returns $\mathsf{Y} \in \mathbb{R}^{h \times w \times d}$. Specifically, $\mathsf{MLP}(\mathsf{X}, c, d)$ does the following:*

- *Collapse the spatial dimensions of the input tensor $\mathsf{X}$ into a matrix $X \in \mathbb{R}^{hw \times c}$.*

- *Compute:*

$$Y_{j,*} = X_{j,*} W + b, \forall j \in [hw]$$

  *where $W \in \mathbb{R}^{c \times d}$ and $b \in \mathbb{R}^{1 \times d}$.*

- *Restore its spatial layout by reshaping $Y$ into $\mathsf{Y} \in \mathbb{R}^{h \times w \times d}$.*

With the definition of MLP, the feedforward layer is defined as follows.

**Definition B.5** (Feedforward Layer). *The feedforward layer* $\mathsf{FFN}(\mathsf{X}, c)$ *takes* $\mathsf{X} \in \mathbb{R}^{h \times w \times c}$ *as input and returns* $\mathsf{Y} \in \mathbb{R}^{h \times w \times c}$. *Specifically,* $\mathsf{FFN}(\mathsf{X}, c)$ *does the following:*

- *Collapse the spatial dimensions of the input tensor* $\mathsf{X}$ *into a matrix* $X \in \mathbb{R}^{hw \times c}$.

- *Compute*
$$Y_{j,*} = X_{j,*} + \mathrm{ReLU}\big(X_{j,*}W_1 + b_1\big)W_2 + b_2, \forall j \in [hw]$$
*where* $W_1, W_2 \in \mathbb{R}^{c \times c}$ *and* $b_1, b_2 \in \mathbb{R}^{1 \times c}$.

- *Restore its spatial layout by reshaping* $Y$ *into* $\mathsf{Y} \in \mathbb{R}^{h \times w \times c}$.

To proceed, the normalization layer is defined as follows.

**Definition B.6** (Layer Normalization Layer). *The layer normalization layer* $\mathsf{LN}(\mathsf{X})$ *takes* $\mathsf{X} \in \mathbb{R}^{h \times w \times c}$ *as input and returns* $\mathsf{Y} \in \mathbb{R}^{h \times w \times c}$. *Specifically,* $\mathsf{LN}(\mathsf{X})$ *does the following:*

- *Collapse the spatial dimensions of the input tensor* $\mathsf{X}$ *into a matrix* $X \in \mathbb{R}^{hw \times c}$.

- *Compute*
$$Y_{j,*} = \frac{X_{j,*} - \mu_j}{\sqrt{\sigma_j^2}}, \forall j \in [hw]$$
*where*
$$\mu_j = \frac{1}{c} \sum_{k=1}^{c} X_{j,k}, \quad \sigma_j^2 = \frac{1}{c} \sum_{k=1}^{c} (X_{j,k} - \mu_j)^2.$$

- *Restore its spatial layout by reshaping* $Y$ *into* $\mathsf{Y} \in \mathbb{R}^{h \times w \times c}$.

We now define the autoregressive transformer used in the HopeFlow model.

**Definition B.7** (Autoregressive Transformer). *Assume we have:*

- Number of scales: $S \in \mathbb{N}$ *is the intermediate steps in HopeFlow.*

- Input tokens: *For each* $i \in [K]$, *the* $\phi_{\mathrm{down}}$ *from Definition B.2 returns* $\mathsf{Y}_i \in \mathbb{R}^{(h/r_i) \times (w/r_i) \times c}$, *where* $r_i = a^{K-i}$ *and* $a \in \mathbb{N}^+$ *is the scaling base.*

- Upsampling function: *For* $i \in [K]$, $\phi_{\mathrm{up},i}(\cdot, a) : \mathbb{R}^{(h/r_i) \times (w/r_i) \times c} \to \mathbb{R}^{(h/r_{i+1}) \times (w/r_{i+1}) \times c}$ *is defined in Definition B.1.*

- Attention layer: *For* $i \in [K]$, $\mathsf{Attn}_i(\cdot) : \mathbb{R}^{(\sum_{j=1}^{i} h/r_j \cdot w/r_j) \times c} \to \mathbb{R}^{(\sum_{j=1}^{i} h/r_j \cdot w/r_j) \times c}$, *as defined in Definition B.3.*

- Feedforward layer: *For* $i \in [K]$, $\mathsf{FFN}_i(\cdot) : \mathbb{R}^{(\sum_{j=1}^{i} h/r_j \cdot w/r_j) \times c} \to \mathbb{R}^{(\sum_{j=1}^{i} h/r_j \cdot w/r_j) \times c}$, *as defined in Definition B.5.*

- Initial condition: $\mathbb{Z}_{\mathrm{init}} \in \mathbb{R}^{(h/r_1) \times (w/r_1) \times c}$ *is the initial token map encoding class information.*

*The autoregressive transformer proceeds as follows:*

- Initialization: *Set* $\mathbb{Z}_1 := \mathbb{Z}_{\mathrm{init}}$.

- Iterative token construction: *Computes*
$$\underset{i}{\mathbb{Z}} := \mathsf{Concat}(\mathsf{Z}_{\mathrm{init}}, \phi_{\mathrm{up},1}(\mathsf{Y}^1, a), \ldots, \phi_{\mathrm{up},i-1}(\mathsf{Y}^{i-1}, a)) \in \mathbb{R}^{(\sum_{j=1}^{i} h/r_j \cdot w/r_j) \times c}, \quad \forall i \geq 2, i \in [K]$$
*where* $\mathsf{Concat}$ *reshapes and concatenates the upsampled tokens into a unified spatial sequence.*

- Transformer block: *Compute:*

$$\mathsf{TF}_i(\underset{i}{\mathbb{Z}}) := \mathsf{FFN_i}(\mathsf{Attn}_i(\underset{i}{\mathbb{Z}})) \in \mathbb{R}^{(\sum_{j=1}^{i} h/r_j \cdot w/r_j) \times c}, \quad \forall i \in [K].$$

- Output extraction: *From $\mathsf{TF}_i(\mathbb{Z}_i)$, extract the last $h/r_i \cdot w/r_i$ rows and reshape to form the output:*

$$\widehat{\mathsf{Y}}_i \in \mathbb{R}^{(h/r_i) \times (w/r_i) \times c}.$$

## B.3 HOPEFLOW ARCHITECTURE

We define the first-order and second-order of the HopeFlow architecture.

**Definition B.8** (First-Order HopeFlow Architecture, formal version of Definition 4.4). *Assume we have:*

- *$S \in \mathbb{N}$ denotes the scale number in HopeFlow, and $i \in [S]$*

- *For a base factor $a \in \mathbb{N}^+$, $r_i := a^{K-i}$ denotes the scale factor.*

- *The interpolation state $\mathsf{F}_i^t \in \mathbb{R}^{(h/r_i) \times (w/r_i) \times c}$ is computed from Definition 4.1.*

- *The end state $\mathsf{F}_i^1 \in \mathbb{R}^{(h/r_i) \times (w/r_i) \times c}$ is computed from downsampling.*

- *The timestep $t_i \in [(i-1)/S, i/S]$.*

- *$\mathsf{Attn}_i(\cdot) : \mathbb{R}^{h/r_i \times w/r_i \times c} \to \mathbb{R}^{h/r_i \times w/r_i \times c}$ is defined in Definition B.3.*

- *$\mathsf{MLP}_i(\cdot, c, d) : \mathbb{R}^{h/r_i \times w/r_i \times c} \to \mathbb{R}^{h/r_i \times w/r_i \times c}$ is defined in Definition B.4.*

- *$\mathsf{LN}_i(\cdot) : \mathbb{R}^{h/r_i \times w/r_i \times c} \to \mathbb{R}^{h/r_i \times w/r_i \times c}$ is defined in Definition B.6.*

*The first-order HopeFlow layers proceeds as follows:*

- Compute time-conditioned parameters:

$$\alpha_1, \alpha_2, \beta_1, \beta_2, \gamma_1, \gamma_2 := \mathsf{MLP}_i(\mathsf{F}_i^1 + t_i \cdot 1_{(h/r_i) \times (w/r_i) \times c}, c, 6c).$$

- Compute intermediate variables:

$$\mathsf{F}_i'^t := \mathsf{Attn}_i(\gamma_1 \circ \mathsf{LN}(\mathsf{F}_i^t) + \beta_1) \circ \alpha_1,$$

*where $\circ$ is the element-wise product.*

- Compute final projection:

$$\mathsf{F}_i''^t := \mathsf{MLP}_i(\gamma_2 \circ \mathsf{LN}(\mathsf{F}_i'^t) + \beta_2, c, c) \circ \alpha_2.$$

*We denote first-order HopeFlow as $\mathsf{F}_i''^t := \mathsf{FlowF}_i(\mathsf{F}_i^1, \mathsf{F}_i^t, t_i)$.*

**Definition B.9** (Second-Order HopeFlow Architecture, formal version of Definition 4.5). *Given the following:*

- *$S \in \mathbb{N}$ denotes the scale number in HopeFlow, and $i \in [S]$*

- *For a base factor $a \in \mathbb{N}^+$, $r_i := a^{K-i}$ denotes the scale factor.*

- *The interpolation state $\mathsf{F}_i^t \in \mathbb{R}^{(h/r_i) \times (w/r_i) \times c}$ is computed from Definition 4.1.*

- *The end state $\mathsf{F}_i^1 \in \mathbb{R}^{(h/r_i) \times (w/r_i) \times c}$ is computed from downsampling.*

- *The timestep $t_i \in [(i-1)/S, i/S]$.*

- *$\mathsf{Attn}_i(\cdot) : \mathbb{R}^{h/r_i \times w/r_i \times c} \to \mathbb{R}^{h/r_i \times w/r_i \times c}$ is defined in Definition B.3.*

- $\mathsf{MLP}_i(\cdot, c, d) : \mathbb{R}^{h/r_i \times w/r_i \times c} \to \mathbb{R}^{h/r_i \times w/r_i \times c}$ *is defined in Definition B.4.*

- $\mathsf{LN}_i(\cdot) : \mathbb{R}^{h/r_i \times w/r_i \times c} \to \mathbb{R}^{h/r_i \times w/r_i \times c}$ *is defined in Definition B.6.*

- $\mathsf{F}_i^{\mathrm{first}} \in \mathbb{R}^{(h/r_i) \times (w/r_i) \times c}$ *denotes the output of the first-order HopeFlow in Defintion B.8.*

*The second-order HopeFlow layers proceeds as follows:*

- Compute time-conditioned parameters:
$$\alpha_1, \alpha_2, \beta_1, \beta_2, \gamma_1, \gamma_2 := \mathsf{MLP}_i(\mathsf{F}_i^1 + t_i \cdot 1_{(h/r_i) \times (w/r_i) \times c}, c, 6c).$$

- Project dimension:
$$\mathsf{F}_i'^t := \mathsf{MLP}_i(\mathsf{Concat}(\mathsf{F}_i^t, \mathsf{F}_i^{\mathrm{first}}), 2, 1).$$

- Compute intermediate variables:
$$\mathsf{F}_i''^t := \mathsf{Attn}_i(\gamma_1 \circ \mathsf{LN}(\mathsf{F}_i'^t) + \beta_1) \circ \alpha_1,$$
where $\circ$ *is the element-wise product.*

- Compute final projection:
$$\mathsf{F}_i'''^t := \mathsf{MLP}_i(\gamma_2 \circ \mathsf{LN}(\mathsf{F}_i''^t) + \beta_2, c, c) \circ \alpha_2.$$

*We denote second-order HopeFlow as $\mathsf{F}_i'''^t := \mathsf{FlowH}_i(\mathsf{F}_i^1, \mathsf{F}_i^t, \mathsf{F}_i^{\mathrm{first}}, t_i)$*

## C   COMPLEXITY OF HOPEFLOW ARCHITECTURE

We analyze the circuit complexity of each module in the HopeFlow architecture in this section. We begin by showing that core operations—such as matrix multiplication, upsampling, and downsampling—are computable in uniform $\mathsf{TC}^0$. We then establish $\mathsf{TC}^0$ implementations for key neural layers including MLP, feedforward, attention, and layer normalization. Finally, we demonstrate that both first-order and second-order HopeFlow layers can be realized in uniform $\mathsf{TC}^0$ with constant depth and polynomial size circuits.

### C.1   COMPUTING MATRIX PRODUCTS IN $\mathsf{TC}^0$

To support later results, we establish that matrix multiplication over floating-point numbers can be efficiently performed within $\mathsf{TC}^0$.

**Lemma C.1** (Matrix Multiplication in $\mathsf{TC}^0$, Lemma B.1 in (Chen et al., 2024)). *Suppose the floating-point precision $p \in O(\mathrm{poly}(n))$, and let $X \in \mathsf{F}_p^{n_1 \times d}$ and $Y \in \mathsf{F}_p^{d \times n_2}$ be two floating-point matrices with dimensions bounded by $n_1, n_2 \in O(\mathrm{poly}(n))$. Then, the $XY$, denoting the matrix product, can be computed by a uniform threshold circuit $\mathsf{TC}^0$ with the following complexity:*

- *Circuit size:* $\mathrm{poly}(n)$,

- *Circuit depth:* $d_{\mathrm{std}} + d_\oplus$,

*where $d_{\mathrm{std}}$ and $d_\oplus$ denote the depths required for basic arithmetic and iterated addition, as defined in Lemma A.9.*

### C.2   COMPUTING DOWN-SAMPLING AND UP-SAMPLING IN $\mathsf{TC}^0$

We now show that nearest-neighbor upsampling can be efficiently computed within the $\mathsf{TC}^0$ complexity class.

**Lemma C.2** (Nearest-Neighbor Upsampling in $\mathsf{TC}^0$). *As defined in Defintion B.1, the nearest-neighbor upsampling function $\phi_{\mathrm{up}}(X, r)$ upsample the input tensor $X \in \mathbb{R}^{h \times w \times c}$ by a scale factor $r \geq 1$. Suppose the precision $p \in O(\mathrm{poly}(n))$, $n = h = w$, $r \leq n$, and $c \leq n$. $\phi_{\mathrm{up}}(X, r) \in \mathsf{TC}^0$ with the following complexity:*

- *Circuit size:* $\text{poly}(n)$,

- *Circuit depth:* $O(1)$.

*Proof.* We show that every output entry is produced in constant depth. For each output index $(i, j, l)$, the upsampling mapping must compute

$$\mathsf{Y}_{i,j,l} = \mathsf{X}_{\lfloor i/r \rfloor, \lfloor j/r \rfloor, l}. \tag{3}$$

First, computing the quotients $\lfloor i/r \rfloor$ and $\lfloor j/r \rfloor$ for $i, j \in [nr]$ is division by the fixed constant $r$, which is known to lie in uniform $\mathsf{TC}^0$ at constant depth and polynomial size. Second, once we have the integer indices $i' = \lfloor i/r \rfloor$ and $j' = \lfloor j/r \rfloor$, wiring the single input value $\mathsf{X}_{i',j',l}$ through to the output is just a multiplexing operation over $\text{poly}(n)$ wires—again realizable in uniform $\mathsf{TC}^0$ at constant depth. Since these two subcircuits (fixed-constant division and unbounded-fan-in multiplexing) both run in parallel for all $(i, j, l)$, the entire nearest-neighbor upsampling is implemented in constant depth and polynomial size. $\square$

We now turn our attention to the downsampling function and show that it can also be computed within the $\mathsf{TC}^0$ complexity class.

**Lemma C.3** (Downsampling in $\mathsf{TC}^0$). *As defined in Definition B.2, the linear downsampling function $\phi_{\text{down}}(\mathsf{X}, r)$ downsample the input tensor $\mathsf{X} \in \mathbb{R}^{h \times w \times c}$ by a scale factor $r \geq 1$. Suppose the precision $p \in O(\text{poly}(n))$, $n = h = w$, $r \leq n$, and $c \leq n$. $\phi_{\text{down}}(\mathsf{X}, r) \in \mathsf{TC}^0$ with the following complexity:*

- *Circuit size:* $\text{poly}(n)$,

- *Circuit depth:* $d_{\text{std}} + d_\oplus$.

*Proof.* From Definition B.2, the down-sampling function is simply a matrix multiplication between a flattened input tensor and a bilinear transformation matrix. The matrix multiplication is in $\mathsf{TC}^0$ by Lemma C.1. $\square$

### C.3 COMPUTING MULTIPLE-LAYER PERCEPTRON IN $\mathsf{TC}^0$

In this subsection, we show the MLP layer lies within the uniform threshold circuit family.

**Lemma C.4** (MLP Computation in $\mathsf{TC}^0$). *As defined in Definition B.4, the $\mathsf{MLP}(\mathsf{X}, c, d)$ takes $\mathsf{X} \in \mathbb{R}^{h \times w \times c}$ as input. Suppose the precision $p \in O(\text{poly}(n))$, $n = h = w$, $r \leq n$, and $c \leq n$. $\mathsf{MLP}(\mathsf{X}, c, d) \in \mathsf{TC}^0$ with the following complexity:*

- *Circuit size:* $\text{poly}(n)$,

- *Circuit depth:* $2d_{\text{std}} + d_\oplus$,

*Proof.* For each row $j \in [hw]$, computing the matrix-vector product $X_{j,*} \cdot W$ requires depth $d_{\text{std}} + d_\oplus$ by Lemma C.1. Adding the bias vector $b$ then requires an additional depth of $d_{\text{std}}$ by Part 1 of Lemma A.9 (basic floating-point addition). Thus, the total depth is $2d_{\text{std}} + d_\oplus$. Since all rows are independent, the circuit depth remains the same, and width is $O(\text{poly}(n))$. $\square$

### C.4 COMPUTING FEED-FORWARD LAYER IN $\mathsf{TC}^0$

In this subsection, we show the feedforward network layer lies within the uniform threshold circuit family.

**Lemma C.5** (FFN Computation in $\mathsf{TC}^0$). *As defined in Definition B.5, the $\mathsf{FFN}(\mathsf{X})$ takes $\mathsf{X} \in \mathbb{R}^{h \times w \times c}$ as input. Suppose the precision $p \in O(\text{poly}(n))$, $n = h = w$, $r \leq n$, and $c \leq n$. $\mathsf{FFN}(\mathsf{X}) \in \mathsf{TC}^0$ with the following complexity:*

- *Circuit size:* $\text{poly}(n)$,

- *Circuit depth:* $6d_{\mathrm{std}} + 2d_\oplus$.

*Proof.* We break down the FFN computation for each $j \in [hw]$ as follows:

- By Lemma C.4, computing the affine transformation $X_{j,*}W_1 + b_1$ requires depth $2d_{\mathrm{std}} + d_\oplus$.

- Applying the ReLU activation $\sigma$ to the result takes an additional depth of $d_{\mathrm{std}}$ by Part 1 of Lemma A.9.

- The next affine transformation $A_1 W_2 + b_2$, where $A_1 = \sigma(X_{j,*}W_1 + b_1)$, also requires depth $2d_{\mathrm{std}} + d_\oplus$ by Lemma C.4.

- Finally, computing the residual connection $X_{j,*} + A_2$ (where $A_2 = A_1 W_2 + b_2$) requires depth $d_{\mathrm{std}}$.

Summing all components gives a total depth of $6d_{\mathrm{std}} + 2d_\oplus$. Since the computation for each $j \in [hw]$ can be performed in parallel, the overall circuit remains within this depth and has polynomial size. $\qquad\square$

## C.5 COMPUTING SINGLE ATTENTION LAYER IN $\mathsf{TC}^0$

In this subsection, we show that a single attention layer lies within the uniform threshold circuit family.

**Lemma C.6** (Attention Layer in $\mathsf{TC}^0$). *As defined in Definition B.3, the $\mathsf{Attn}(\mathsf{X})$ takes $\mathsf{X} \in \mathbb{R}^{h \times w \times c}$ as input. Suppose the precision $p \in O(\mathrm{poly}(n))$, $n = h = w$, $r \le n$, and $c \le n$. $\mathsf{Attn}(\mathsf{X}) \in \mathsf{TC}^0$ with the following complexity:*

- *Circuit size:* $\mathrm{poly}(n)$,

- *Circuit depth:* $6(d_{\mathrm{std}} + d_\oplus) + d_{\exp}$.

*Proof.* We decompose the attention layer into several computational stages:

- *Key-Query Product:* The term $W_Q W_K^\top$ is precomputed and fixed. The matrix-vector product $X_{i,*}W_Q W_K^\top X_{j,*}^\top$ requires two applications of matrix multiplication, yielding depth $2(d_{\mathrm{std}} + d_\oplus)$ by Lemma C.1.

- *Score Computation and Exponentiation:* Computing each pairwise attention score $s_{i,j}$ as above, followed by computing $A_{i,j} = \exp(s_{i,j})$, adds $d_{\exp}$ to the total depth. Hence, the full attention matrix $A$ can be computed with depth $3(d_{\mathrm{std}} + d_\oplus) + d_{\exp}$.

Next, we perform the normalization and projection steps:

- *Row Normalization:* Computing the row-wise sums $D = \mathrm{diag}(A\mathbf{1}_n)$ requires depth $d_\oplus$; inverting the diagonal matrix $D$ requires depth $d_{\mathrm{std}}$.

- *Value Projection:* Computing $AXW_V$ (matrix multiplication followed by linear projection) requires depth $2(d_{\mathrm{std}} + d_\oplus)$. Multiplying with $D^{-1}$ adds an additional depth of $d_{\mathrm{std}}$.

Summing all contributions gives the total depth is $6(d_{\mathrm{std}} + d_\oplus) + d_{\exp}$, and the total circuit size remains $\mathrm{poly}(n)$. $\qquad\square$

### C.6 COMPUTING LAYER-WISE NORM LAYER IN $\mathsf{TC}^0$

In this subsection, we show that a layer normalization layer lies within the uniform threshold circuit family.

**Lemma C.7** (Layer Normalization in $\mathsf{TC}^0$)**.** *As defined in Definition B.6, the* $\mathsf{LN}(\mathsf{X})$ *takes* $\mathsf{X} \in \mathbb{R}^{h \times w \times c}$ *as input. Suppose the precision* $p \in O(\mathrm{poly}(n))$, $n = h = w$, $r \leq n$, *and* $c \leq n$. $\mathsf{LN}(\mathsf{X}) \in \mathsf{TC}^0$ *with the following complexity:*

- *Circuit size:* $\mathrm{poly}(n)$,

- *Circuit depth:* $5d_{\mathrm{std}} + 2d_{\oplus} + d_{\mathrm{sqrt}}$.

*Proof.* The computation of $\mathsf{LN}(\mathsf{X})$ involves the following components for each $j \in [hw]$:

- *Mean computation:* As shown in Lemma A.9, calculating $\mu_j = \frac{1}{c} \sum_{k=1}^{c} X_{j,k}$ requires depth $d_{\mathrm{std}} + d_{\oplus}$.

- *Variance computation:* Calculating $\sigma_j^2 = \frac{1}{c} \sum_{k=1}^{c} (X_{j,k} - \mu_j)^2$ requires two additional applications of floating-point operations, yielding depth $2d_{\mathrm{std}} + d_{\oplus}$.

- *Normalization:* Computing the normalized output $Y_{j,*} = \frac{X_{j,*} - \mu_j}{\sqrt{\sigma_j^2}}$ requires subtracting the mean and dividing by the square root of the variance. This adds another depth of $2d_{\mathrm{std}} + d_{\oplus} + d_{\mathrm{sqrt}}$ by Lemmas A.9 and A.11.

Summing all contributions, the total circuit depth is $5d_{\mathrm{std}} + 2d_{\oplus} + d_{\mathrm{sqrt}}$, and the total size remains $\mathrm{poly}(n)$. $\square$

### C.7 COMPUTING FIRST-ORDER HOPEFLOW LAYER IN $\mathsf{TC}^0$

In this subsection, we show that the first-order HopeFlow layer lies within the uniform threshold circuit family.

**Lemma C.8** (First-order HopeFlow layer computation in $\mathsf{TC}^0$, formal version of Lemma 5.1)**.** *As defined in Definition B.8, the* $\mathsf{FlowF}(\mathsf{X})$ *takes* $\mathsf{X} \in \mathbb{R}^{h \times w \times c}$ *as input. Suppose the precision* $p \in O(\mathrm{poly}(n))$, $n = h = w$, $r \leq n$, *and* $c \leq n$. $\mathsf{FlowF}(\mathsf{X}) \in \mathsf{TC}^0$ *with the following complexity:*

- *Circuit size:* $\mathrm{poly}(n)$.

- *Circuit depth:* $26d_{\mathrm{std}} + 12d_{\oplus} + 2d_{\mathrm{sqrt}} + d_{\exp}$.

*Proof.* The first step of first-order HopeFlow is a MLP and by Lemma C.4, it is in a $\mathsf{TC}^0$ family with depth $2d_{\mathrm{std}} + d_{\oplus}$ and size of $\mathrm{poly}(n)$.

The second step is a layer normalization layer and by Lemma C.7, $\mathsf{LN}(\mathsf{F}_i^t)$ is in a $\mathsf{TC}^0$ family with depth $5d_{\mathrm{std}} + 2d_{\oplus} + d_{\mathrm{sqrt}}$. By Lemma A.9, $A_1 = \gamma_1 \circ \mathsf{LN}(\mathsf{F}_t) + \beta_1$ is in a $\mathsf{TC}^0$ family with depth $2d_{\mathrm{std}}$. By Lemma C.6, $A_2 = \mathsf{Attn}(A_1)$ is in a $\mathsf{TC}^0$ family with depth $6(d_{\mathrm{std}} + d_{\oplus}) + d_{\exp}$. By Lemma A.9 again, scaling $A_2 \circ \alpha_1$ is in a $\mathsf{TC}^0$ family with depth $d_{\mathrm{std}}$. The total depth requires $14d_{\mathrm{std}} + 8d_{\oplus} + d_{\mathrm{sqrt}} + d_{\exp}$ for step 2.

The third step is a layer normalization layer and by Lemma C.7, $\mathsf{LN}(\mathsf{F}_i'^t)$ is in a $\mathsf{TC}^0$ family with depth $5d_{\mathrm{std}} + 2d_{\oplus} + d_{\mathrm{sqrt}}$. By Lemma A.9, $A_3 = \gamma_2 \circ \mathsf{LN}(\mathsf{F}_i'^t) + \beta_2$ is in a $\mathsf{TC}^0$ family with depth $2d_{\mathrm{std}}$. By Lemma C.4, $A_4 = \mathsf{MLP}(A_3, c, c)$ is in a $\mathsf{TC}^0$ family with depth $2d_{\mathrm{std}} + d_{\oplus}$. By Lemma A.9 again, $A_4 \circ \alpha_2$ requires depth $d_{\mathrm{std}}$. The total depth requires $10d_{\mathrm{std}} + 3d_{\oplus} + d_{\mathrm{sqrt}}$ for step 3.

In summary, $\mathsf{FlowF}(\mathsf{X})$ is in a $\mathsf{TC}^0$ family with depth $26d_{\mathrm{std}} + 12d_{\oplus} + 2d_{\mathrm{sqrt}} + d_{\exp}$ and size $\mathrm{poly}(n)$.

$\square$

### C.8 COMPUTING SECOND-ORDER HOPEFLOW LAYER IN $\mathsf{TC}^0$

In this subsection, we show that the second-order HopeFlow layer lies within the uniform threshold circuit family.

**Lemma C.9** (Second-order HopeFlow layer computation in $\mathsf{TC}^0$, formal version of Lemma 5.2)**.** *As defined in Definition B.9, the* $\mathsf{FlowH}(\mathsf{X})$ *takes* $\mathsf{X} \in \mathbb{R}^{h \times w \times c}$ *as input. Suppose the precision* $p \in O(\mathrm{poly}(n))$, $n = h = w$, $r \leq n$, *and* $c \leq n$. $\mathsf{FlowH}(\mathsf{X}) \in \mathsf{TC}^0$ *with the following complexity:*

- *Circuit size:* $\mathrm{poly}(n)$.

- *Circuit depth:* $28d_{\mathrm{std}} + 13d_{\oplus} + 2d_{\mathrm{sqrt}} + d_{\mathrm{exp}}$.

*Proof.* The first step of second-order HopeFlow is a MLP and by Lemma C.4, it is in a $\mathsf{TC}^0$ family with depth $2d_{\mathrm{std}} + d_{\oplus}$ and size of $\mathrm{poly}(n)$.

The second step is a layer normalization layer and by Lemma C.7, $\mathsf{LN}(\mathsf{F}_i^t)$ is in a $\mathsf{TC}^0$ family with depth $5d_{\mathrm{std}} + 2d_{\oplus} + d_{\mathrm{sqrt}}$. By Lemma A.9, $A_1 = \gamma_1 \circ \mathsf{LN}(\mathsf{F}_t) + \beta_1$ is in a $\mathsf{TC}^0$ family with depth $2d_{\mathrm{std}}$. By Lemma C.6, $A_2 = \mathsf{Attn}(A_1)$ is in a $\mathsf{TC}^0$ family with depth $6(d_{\mathrm{std}} + d_{\oplus}) + d_{\mathrm{exp}}$. By Lemma A.9 again, scaling $A_2 \circ \alpha_1$ is in a $\mathsf{TC}^0$ family with depth $d_{\mathrm{std}}$. The total depth requires $14d_{\mathrm{std}} + 8d_{\oplus} + d_{\mathrm{sqrt}} + d_{\mathrm{exp}}$ for step 2.

The third step is a layer normalization layer and by Lemma C.7, $\mathsf{LN}(\mathsf{F}_i'^t)$ is in a $\mathsf{TC}^0$ family with depth $5d_{\mathrm{std}} + 2d_{\oplus} + d_{\mathrm{sqrt}}$. By Lemma A.9, $A_3 = \gamma_2 \circ \mathsf{LN}(\mathsf{F}_i'^t) + \beta_2$ is in a $\mathsf{TC}^0$ family with depth $2d_{\mathrm{std}}$. By Lemma C.4, $A_4 = \mathsf{MLP}(A_3, c, c)$ is in a $\mathsf{TC}^0$ family with depth $2d_{\mathrm{std}} + d_{\oplus}$. By Lemma A.9 again, $A_4 \circ \alpha_2$ requires depth $d_{\mathrm{std}}$. The total depth requires $10d_{\mathrm{std}} + 3d_{\oplus} + d_{\mathrm{sqrt}}$ for step 3.

In the last step, by Lemma C.7, $\mathsf{LN}(\mathsf{F}_i''^t)$ is in a $\mathsf{TC}^0$ family with depth $5d_{\mathrm{std}} + 2d_{\oplus} + d_{\mathrm{sqrt}}$. By Lemma A.9, $A_3 = \gamma_2 \circ \mathsf{LN}(\mathsf{F}_i''^t) + \beta_2$ is in a $\mathsf{TC}^0$ family with depth $2d_{\mathrm{std}}$. By Lemma C.4, $A_4 = \mathsf{MLP}(A_3, c, c)$ is in a $\mathsf{TC}^0$ family with depth $2d_{\mathrm{std}} + d_{\oplus}$. By Lemma A.9 again, $A_4 \circ \alpha_2$ is in a $\mathsf{TC}^0$ family with depth $d_{\mathrm{std}}$. The total depth is $10d_{\mathrm{std}} + 3d_{\oplus} + d_{\mathrm{sqrt}}$.

In summary, $\mathsf{FlowH}(\mathsf{X})$ is in a $\mathsf{TC}^0$ family with depth $28d_{\mathrm{std}} + 13d_{\oplus} + 2d_{\mathrm{sqrt}} + d_{\mathrm{exp}}$ and size $\mathrm{poly}(n)$ to simulate the second-order HopeFlow layer.

$\square$

## D STATISTICAL CONVERGENCE GUARANTEES OF HOPEFLOW

This section establishes the statistical convergence guarantees for the HopeFlow architecture. We begin by introducing the necessary mathematical background, including the modulus of smoothness and Besov spaces (Section D.1). We then define key time partitioning variables used to analyze the convergence over different regimes. In Section D.2, we present error bounds for first-order flow matching in both small-time and large-time regimes. Finally, Section D.3 lists the technical assumptions on the data distribution and flow parameters required for the convergence results to hold.

### D.1 BESOV SPACE

To quantify the smoothness of a function, we use the $r$-th modulus of smoothness.

**Definition D.1** ($r$-th Modulus of Smoothness, Definition 2.2 on Page 3 in (Oko et al., 2023))**.** *Let* $p \in (0, \infty]$ *and let* $f \in L^p(\Omega)$. *The* $r$-th *modulus of smoothness of* $f$ *is defined by:*

$$w_{r,p}(f, t) := \sup_{\|h\|_2 \leq t} \|\Delta_h^r(f)\|_p,$$

*where the* $r$-th *order difference operator* $\Delta_h^r(f)$ *is given by:*

$$\Delta_h^r(f)(x) := \begin{cases} \sum_{j=0}^r \binom{r}{j} \cdot (-1)^{r-j} \cdot f(x + jh) & \text{if } x + jh \in \Omega \text{ for all } j; \\ 0 & \text{otherwise.} \end{cases}$$

With the definition of the modulus of smoothness in place, we now introduce the Besov space $B_{p,q}^s(\Omega)$, which provides a more nuanced characterization of function smoothness.

**Definition D.2** (Besov Space $B_{p,q}^s(\Omega)$, Definition 2.3 on page 3 in (Oko et al., 2023))**.** *Let the following parameters be given:*

- *$p > 0$, $q \le \infty$, and $s > 0$,*

- *Let $r := \lfloor s \rfloor + 1$,*

- *Let $w_{r,p}(f, t)$ denote the $r$-th modulus of smoothness of $f$, as defined in Definition D.1.*

*Then the Besov space $B_{p,q}^s(\Omega)$ is defined as the set:*

$$B_{p,q}^s := \{f \in L^p(\Omega) \mid \|f\|_{B_{p,q}^s} < \infty\},$$

*where*

$$|f|_{B_{p,q}^s} = \begin{cases} (\int_0^\infty (t^{-s} w_{r,p}(f, t))^q \frac{\mathrm{d}t}{t})^{\frac{1}{q}} & \text{if } q < \infty; \\ \sup_{t>0} \{t^{-s} w_{r,p}(f, t)\} & \text{if } q = \infty, \end{cases}$$

*is the Besov seminorm and*

$$\|f\|_{B_{p,q}^s} := \|f\|_p + |f|_{B_{p,q}^s}$$

*is the full norm.*

To facilitate the analysis, we partition the time horizon into regimes where different approximation arguments apply. The following definition introduces key time thresholds and dyadic steps used to control the behavior of $\alpha(t)$, $\beta(t)$, and their derivatives.

**Definition D.3** (Time Variables and Partition, Definition 5.10 on page 11 in (Gong et al., 2025))**.** *We define the following time-related variables:*

- *Initial time: $T_0 := N^{-R_0}$.*

- *Intermediate threshold: $T_* := N^{-(\kappa^{-1}-\delta)/d}$.*

- *Boundary time: $t_{j_*} \in [T_*, 3T_*]$ denotes a critical transition point where different generalization bounds are applied.*

- *Dyadic sequence: For $j \in [K]$, define $t_j := 2t_{j-1}$, with the base case $t_0 := T_0$ and the final value $t_K := 1$.*

### D.2 FIRST ORDER ERROR BOUND

We now present a preliminary result to characterize the error bound in first-order flow matching. The following definition introduces the form of the interpolated vector field and its derivatives.

**Definition D.4** (Interpolated Vector Field and Derivatives)**.** *Let $x_{1,0}$ and $x_{1,1}$ denote the initial and target distributions, respectively. Define the time-dependent vector field $x_{1,t}$ as:*

$$x_{1,t} := \alpha_t x_{1,0} + \beta_t x_{1,1},$$

*where $\alpha_t$ and $\beta_t$ are time-dependent interpolation functions. The first and second time derivatives of $x_{1,t}$ are given by:*

$$x_{1,t}' = \alpha_t' x_{1,0} + \beta_t' x_{1,1},$$
$$x_{1,t}'' = \alpha_t'' x_{1,0} + \beta_t'' x_{1,1}.$$

We now present two approximation results that provide error bounds for first-order flow matching, depending on the time regime.

**Lemma D.5** (Theorem 7 in (Fukumizu et al., 2025))**.** *Suppose the following conditions hold:*

- *Assumptions D.7, D.8, D.9, D.10, D.12, and D.14 are satisfied.*

- $\alpha_t$ and $\beta_t$ are defined as in Definition D.4.

- $C_6$ is a constant independent of $t$.

Then, for sufficiently large $N$, there exists a neural network $\phi_1 \in \mathcal{M}(L, W, S, B)$ such that

$$\int \|\phi_1(x_1, t) - v_t(x_1)\|_2^2 \cdot p_t(x_1) \, dx_1 \leq C_6 \cdot \left(\alpha_t'^2 \log N + \beta_t'^2\right) \cdot N^{-2s/d},$$

for all $t \in [T_0, 3T_*]$, where the network parameters satisfy:

$$L = O(\log^4 N), \quad \|W\|_\infty = O(N \log^6 N), \quad S = O(N \log^8 N), \quad B = \exp(O(\log N \log \log N)).$$

**Lemma D.6** (Theorem 8 in (Fukumizu et al., 2025)). *Suppose the following conditions hold:*

- *Fix any $t_* \in [T_*, 1]$ and arbitrary $\eta > 0$.*

- *Assumptions D.7, D.8, D.9, D.10, D.12, and D.14 are satisfied.*

- *$\alpha_t$ and $\beta_t$ are defined as in Definition D.4.*

- *$C_7 > 0$ is a constant independent of $t$.*

*Then there exists a neural network $\phi_2 \in \mathcal{M}(L, W, S, B)$ such that*

$$\int \|\phi_2(x_1, t) - v_t(x_1)\|_2^2 \cdot p_t(x_1) \, dx_1 \leq C_7 \cdot \left(\alpha_t'^2 \log N + \beta_t'^2\right) \cdot N^{-\eta},$$

*for all $t \in [2t_*, 1]$, where the network parameters satisfy:*

$$L = O(\log^4 N), \quad \|W\|_\infty = O(N), \quad S = O(t_*^{-d\kappa} N^{\delta\kappa}), \quad B = \exp(O(\log N \log \log N)).$$

### D.3 BASIC ASSUMPTIONS

Our analysis relies on the following assumptions regarding the target probability distribution $P_0$.

**Assumption D.7** (Target Distribution Regularity, Assumption 5.2 in (Gong et al., 2025)). *Let $I_N^d$ denote the contracted cube defined by*

$$I_N^d := \left[-1 + N^{-(1-\kappa\delta)}, \, 1 - N^{-(1-\kappa\delta)}\right]^d,$$

*where $N$ is the sample size and the parameters $\kappa$ and $\delta$ satisfy Assumption D.9. We assume that the target probability distribution $P_0$ has support on $I^d$ and that its density $p_0$ satisfies:*

- *$p_0 \in B_{p',q'}^s(I^d)$;*

- *$p_0 \in B_{p',q'}^{\check{s}}(I^d \setminus I_N^d)$ with $\check{s} > \max\{6s, 1\}$.*

**Assumption D.8** (Density Bounds, Assumption 5.3 in (Gong et al., 2025)). *There exists a constant $C_0 > 0$ such that the target density $p_0$ satisfies the uniform bounds:*

$$C_0^{-1} \leq p_0(x_1) \leq C_0, \quad \forall x_1 \in I^d.$$

**Assumption D.9** (Interpolation Function Parametrization, Assumption 5.4 in (Gong et al., 2025)). *Let $\kappa \geq 1/2$, $b_0 > 0$, $\widetilde{\kappa} > 0$, and $\widetilde{b}_0 > 0$. For sufficiently small $t \geq T_0$, we assume:*

$$\alpha_t = b_0 t^\kappa, \qquad 1 - \beta_t = \widetilde{b}_0 t^{\widetilde{\kappa}}.$$

*Additionally, there exists a constant $D_0 > 0$ such that:*

$$D_0^{-1} \leq \alpha_t^2 + \beta_t^2 \leq D_0, \qquad \forall t \in [T_0, 1].$$

**Assumption D.10** (First-Order Derivative Bounds, Assumption 5.5 in (Gong et al., 2025)). *$\exists K_0 > 0$ and $K_0$ is a constant such that the first derivatives of $\alpha_t$ and $\beta_t$ satisfy:*

$$|\alpha_t'| + |\beta_t'| \leq N^{K_0}, \qquad \forall t \in [T_0, 1].$$

**Assumption D.11** (Second-Order Derivative Bounds, Assumption 5.6 in (Gong et al., 2025)). $\exists K_0 > 0$ and $K_0$ is a constant such that the second derivatives of $\alpha_t$ and $\beta_t$ satisfy:

$$|\alpha_t''| + |\beta_t''| \leq N^{K_0}, \qquad \forall t \in [T_0, 1].$$

**Assumption D.12** (Integral Bound on First Derivatives for $\kappa = 1/2$, Assumption 5.7 in (Gong et al., 2025)). Let $s$ be the smoothness parameter from Definition D.2, and let $\kappa$ satisfy Assumption D.9. Let $T_0$ be defined as in Definition D.3, and fix $R_0 \geq \frac{s+1}{\min\{\kappa, \bar{\kappa}\}}$.

If $\kappa = 1/2$, then there exist constants $b_1 > 0$ and $D_1 > 0$ such that, for any $0 \leq \gamma < R_0$, the following bound holds:

$$\int_{T_0}^{N^{-\gamma}} \left( \alpha_t'^2 + \beta_t'^2 \right) dt \leq D_1 \cdot \log^{b_1} N.$$

**Assumption D.13** (Integral Bound on Second Derivatives for $\kappa = 1/2$, Assumption 5.8 in (Gong et al., 2025)). Under the same setting as Assumption D.12, there exist constants $b_1 > 0$ and $D_1 > 0$ such that, for any $0 \leq \gamma < R_0$, the following bound holds:

$$\int_{T_0}^{N^{-\gamma}} \left( \alpha_t''^2 + \beta_t''^2 \right) dt \leq D_1 \cdot \log^{b_1} N.$$

**Assumption D.14** (Bounded Derivative of Conditional Mean, Assumption 5.9 in (Gong et al., 2025)). There exists a constant $C_L > 0$ such that, for all $t \in [T_0, 1]$,

$$\left\| \frac{d}{dx_1} \int y \, p_t(y \mid x_1) \, dy \right\| \leq C_L.$$

# E    PROVABLY EFFICIENT CRITERIA

This section introduces the Fast HopeFlow architecture and establishes its provable efficiency. Section E.1 formally defines the first- and second-order variants of Fast HopeFlow, which incorporate approximate attention to reduce computational complexity. Section E.2 analyzes the inference runtime of the original HopeFlow architecture, demonstrating a baseline complexity of $O(n^{4+o(1)})$. Section E.3 presents the runtime analysis for Fast HopeFlow and proves a near-quadratic runtime of $O(n^{2+o(1)})$ using approximate attention mechanisms.

## E.1    FAST HOPEFLOW ARCHITECTURE

We define the first-order and second-order of the Fast HopeFlow architecture.

**Definition E.1** (First-Order Fast HopeFlow Architecture). Assume we have:

- $S \in \mathbb{N}$ denotes the scale number in Fast HopeFlow, and $i \in [S]$

- For a base factor $a \in \mathbb{N}^+$, $r_i := a^{K-i}$ denotes the scale factor.

- The interpolation state $\mathsf{F}_i^t \in \mathbb{R}^{(h/r_i) \times (w/r_i) \times c}$ is computed from Definition 4.1.

- The end state $\mathsf{F}_i^1 \in \mathbb{R}^{(h/r_i) \times (w/r_i) \times c}$ is computed from downsampling.

- The timestep $t_i \in [(i-1)/S, i/S]$.

- $\mathsf{AAttnC}_i(\cdot) : \mathbb{R}^{h/r_i \times w/r_i \times c} \to \mathbb{R}^{h/r_i \times w/r_i \times c}$ is defined in Definition 7.1.

- $\mathsf{MLP}_i(\cdot, c, d) : \mathbb{R}^{h/r_i \times w/r_i \times c} \to \mathbb{R}^{h/r_i \times w/r_i \times c}$ is defined in Definition B.4.

- $\mathsf{LN}_i(\cdot) : \mathbb{R}^{h/r_i \times w/r_i \times c} \to \mathbb{R}^{h/r_i \times w/r_i \times c}$ is defined in Definition B.6.

The first-order Fast HopeFlow layers proceeds as follows:

- Compute time-conditioned parameters:

$$\alpha_1, \alpha_2, \beta_1, \beta_2, \gamma_1, \gamma_2 := \mathsf{MLP}_i(\mathsf{F}_i^1 + t_i \cdot 1_{(h/r_i) \times (w/r_i) \times c}, c, 6c).$$

- Compute intermediate variables:

$$\mathsf{F}_i'^t := \mathsf{AAttnC}_i(\gamma_1 \circ \mathsf{LN}(\mathsf{F}_i^t) + \beta_1) \circ \alpha_1,$$

  *where $\circ$ is the element-wise product.*

- Compute final projection:

$$\mathsf{F}_i''^t := \mathsf{MLP}_i(\gamma_2 \circ \mathsf{LN}(\mathsf{F}_i'^t) + \beta_2, c, c) \circ \alpha_2.$$

*We denote first-order Fast HopeFlow as $\mathsf{F}_i''^t := \mathsf{FFlowF}_i(\mathsf{F}_i^1, \mathsf{F}_i^t, t_i)$ .*

**Definition E.2** (Second-Order Fast HopeFlow Architecture). *Given the following:*

- *$S \in \mathbb{N}$ denotes the scale number in Fast HopeFlow, and $i \in [S]$*

- *For a base factor $a \in \mathbb{N}^+$, $r_i := a^{K-i}$ denotes the scale factor.*

- *The interpolation state $\mathsf{F}_i^t \in \mathbb{R}^{(h/r_i) \times (w/r_i) \times c}$ is computed from Definition 4.1.*

- *The end state $\mathsf{F}_i^1 \in \mathbb{R}^{(h/r_i) \times (w/r_i) \times c}$ is computed from downsampling.*

- *The timestep $t_i \in [(i-1)/S, i/S]$.*

- *$\mathsf{AAttnC}_i(\cdot) : \mathbb{R}^{h/r_i \times w/r_i \times c} \to \mathbb{R}^{h/r_i \times w/r_i \times c}$ is defined in Definition 7.1.*

- *$\mathsf{MLP}_i(\cdot, c, d) : \mathbb{R}^{h/r_i \times w/r_i \times c} \to \mathbb{R}^{h/r_i \times w/r_i \times c}$ is defined in Definition B.4.*

- *$\mathsf{LN}_i(\cdot) : \mathbb{R}^{h/r_i \times w/r_i \times c} \to \mathbb{R}^{h/r_i \times w/r_i \times c}$ is defined in Definition B.6.*

- *$\mathsf{F}_i^{\text{first}} \in \mathbb{R}^{(h/r_i) \times (w/r_i) \times c}$ denotes the output of the first-order Fast HopeFlow in Definition B.8.*

*The second-order Fast HopeFlow layers proceeds as follows:*

- Compute time-conditioned parameters:

$$\alpha_1, \alpha_2, \beta_1, \beta_2, \gamma_1, \gamma_2 := \mathsf{MLP}_i(\mathsf{F}_i^1 + t_i \cdot 1_{(h/r_i) \times (w/r_i) \times c}, c, 6c).$$

- Project dimension:

$$\mathsf{F}_i'^t := \mathsf{MLP}_i(\mathsf{Concat}(\mathsf{F}_i^t, \mathsf{F}_i^{\text{first}}), 2, 1).$$

- Compute intermediate variables:

$$\mathsf{F}_i''^t := \mathsf{AAttnC}_i(\gamma_1 \circ \mathsf{LN}(\mathsf{F}_i'^t) + \beta_1) \circ \alpha_1,$$

  *where $\circ$ is the element-wise product.*

- Compute final projection:

$$\mathsf{F}_i'''^t := \mathsf{MLP}_i(\gamma_2 \circ \mathsf{LN}(\mathsf{F}_i''^t) + \beta_2, c, c) \circ \alpha_2.$$

*We denote second-order Fast HopeFlow as $\mathsf{F}_i'''^t := \mathsf{FFlowH}_i(\mathsf{F}_i^1, \mathsf{F}_i^t, \mathsf{F}_i^{\text{first}}, t_i)$*

E.2 RUNTIME ANALYSIS OF THE ORIGINAL HOPEFLOW INFERENCE PIPELINE

In this subsection, we give a runtime analysis about the HopeFlow inference pipeline.

**Lemma E.3** (Inference Runtime of Original HopeFlow Architecture, formal version of Lemma 7.3). *Given the following:*

- *$\mathsf{X} \in \mathbb{R}^{h \times w \times c}$ is the input tensor.*

- *$K = O(1)$ is the number of scales, and $i \in [K]$.*

- *For a base factor $a \in \mathbb{N}^+$, $r_i := a^{K-i}$ is the scale factor.*

- *$\phi_{\text{up},i}(\cdot, a)$ is the upsampling function from Definition B.1.*

- *$\text{Attn}_i(\cdot)$ is the attention layer from Definition B.3.*

- *$\text{FFN}_i(\cdot)$ is the feed forward layer from Definition B.5.*

- *$\text{FlowF}_i(\cdot, \cdot, \cdot)$ is the first-order HopeFlow layer from Definition B.8.*

- *$\text{FlowH}_i(\cdot, \cdot, \cdot, \cdot)$ is the second-order HopeFlow layer from Definition B.9.*

*HopeFlow achieves inference in $O(n^{4+o(1)})$ time.*

*Proof.* We analyze the runtime of upsampling layers, first-order HopeFlow layer, and second-order HopeFlow layer respective.

We start with the upsampling layers. Each layer $i$ in the HopeFlow model contains $\phi_{\text{up},1}(\cdot, 2), \ldots, \phi_{\text{up},i-1}(\cdot, 2)$ upsampling functions. The $\phi_{\text{up},i-1}(\cdot, 2)$ function runs in $O(n^2 c / 2^{2(K-i)})$ time, and the total runtime of all upsampling function in layer $i$ is $O(n^2 c \cdot \frac{1}{2^{2K}} \cdot (1 - \frac{1}{4^i}))$. Summarizing all layers, the total runtime of upsampling functions the HopeFlow model is:

$$\mathcal{T}_{\text{up}} = \sum_{i=1}^{K} O(n^2 c \cdot \frac{1}{2^{2K}} \cdot (1 - \frac{1}{4^i}))$$
$$= O(n^{2+o(1)}).$$

Then, we analyze the runtime of first-order HopeFlow layer. The input tensor for each layer $i$ is $(n/2^{K-i}) \times (n/2^{K-i}) \times c$, and the runtime of each layer is dominated by the attention layer that is $O(n^4 c / 2^{4(K-i)})$. Summarizing all layers, the total runtime of first-order HopeFlow is

$$\mathcal{T}_{\text{FlowF}} = \sum_{i=1}^{K} O(n^4 c / 2^{4(K-i)})$$
$$= O(n^{4+o(1)}).$$

Lastly, we analyze the runtime of second-order HopeFlow layer. The input tensor for each layer $i$ is $(n/2^{K-i}) \times (n/2^{K-i}) \times c$, and the runtime of each layer is also dominated by the attention layer that is $O(n^4 c / 2^{4(K-i)})$. Summarizing all layers, the total runtime of second-order HopeFlow is

$$\mathcal{T}_{\text{FlowH}} = \sum_{i=1}^{K} O(n^4 c / 2^{4(K-i)})$$
$$= O(n^{4+o(1)}).$$

Hence, the total runtime of HopeFlow architecture is

$$\mathcal{T}_{\text{ori}} = \mathcal{T}_{\text{up}} + \mathcal{T}_{\text{FlowF}} + \mathcal{T}_{\text{FlowH}}$$
$$= O(n^{4+o(1)}).$$

$\square$

### E.3 RUNTIME ANALYSIS OF THE FAST HOPEFLOW INFERENCE PIPELINE

In this subsection, we give a runtime analysis about the Fast HopeFlow inference pipeline.

**Lemma E.4** (Inference Runtime of Fast HopeFlow Architecture). *Given the following:*

- *$\mathsf{X} \in \mathbb{R}^{h \times w \times c}$ is the input tensor.*

- *$K = O(1)$ is the number of scales, and $i \in [K]$.*

- *For a base factor $a \in \mathbb{N}^+$, $r_i := a^{K-i}$ is the scale factor.*

- *$\phi_{\mathrm{up},i}(\cdot, a)$ is the upsampling function from Definition B.1.*

- *$\mathsf{AAttnC}_i(\cdot)$ is the attention layer from Definition 7.1.*

- *$\mathsf{FFN}_i(\cdot)$ is the feed forward layer from Definition B.5.*

- *$\mathsf{FFlowF}_i(\cdot, \cdot, \cdot)$ is the first-order Fast HopeFlow layer from Definition E.1.*

- *$\mathsf{FFlowH}_i(\cdot, \cdot, \cdot, \cdot)$ is the second-order Fast HopeFlow layer from Definition E.2.*

*Fast HopeFlow achieves inference in $O(n^{2+o(1)})$ time.*

*Proof.* The total runtime of upsampling functions remain $O(n^{2+o(1)})$. As we replace the attention layer with the approximate attention, the runtime of each first-order Fast HopeFlow layer is dominated by MLP which is $O(n^{2+o(1)})$, and $\mathcal{T}_{\mathrm{FFlowF}} = \sum_{i=1}^{K} O(n^{2+o(1)}) = O(n^{2+o(1)})$. Similarly, the second-order Fast HopeFlow layer is $\mathcal{T}_{\mathrm{FFlowF}} = \sum_{i=1}^{K} O(n^{2+o(1)}) = O(n^{2+o(1)})$. Hence $\mathcal{T}_{\mathrm{fast}} = \mathcal{T}_{\mathrm{up}} + \mathcal{T}_{\mathrm{FFlowF}} + \mathcal{T}_{\mathrm{FFlowH}} = O(n^{2+o(1)})$.

$\square$

## LLM USAGE DISCLOSURE

LLMs were used only to polish language, such as grammar and wording. These models did not contribute to idea creation or writing, and the authors take full responsibility for this paper's content.

