# OpenReview forum: "Provably Efficient High-Order Flow Matching in Pixel Space"
_ICLR.cc/2026/Conference — ICLR 2026 Conference Withdrawn Submission_

### Official Review · Reviewer_252L · 2025-10-25

**Soundness:** 1
**Presentation:** 1
**Contribution:** 1
**Rating:** 2
**Confidence:** 5

**Summary:**

HopeFlow is the  cascade flow model that jointly learns both pixel-space velocity and acceleration fields in an end-to-end manner. By incorporating second-order dynamics, it better aligns mid-horizon dependencies and high-curvature regions, leading to smoother and more stable transport trajectories during image generation.

**Strengths:**

The paper provides a rigorous theoretical framework for high-order flow matching in pixel space, extending prior first-order methods. The related work is well-organized and sufficient to position the approach within existing diffusion and flow-based generative modeling literature.

**Weaknesses:**

A major weakness of this paper is the complete absence of empirical experiments or quantitative evaluations. Without any empirical evidence, it is impossible to assess whether the proposed theoretical framework offers practical benefits over existing methods.

**Questions:**

See the Weaknesses section regarding the absence of experiments and empirical evidence.

---

> ### Author Response · Authors · 2025-11-27
>
> Thank you for your helpful feedback. We will update the paper in the next revision.

---

### Official Review · Reviewer_G4sF · 2025-10-31

**Soundness:** 2
**Presentation:** 1
**Contribution:** 2
**Rating:** 2
**Confidence:** 3

**Summary:**

This paper’s contribution lies in proposing a model architecture that learns both velocity and acceleration fields for image generation. The authors provide formal proofs showing that the proposed architecture belongs to the uniform TC⁰ circuit complexity class, suggesting provable efficiency and parallelizability.

**Strengths:**

The proposed idea of incorporating acceleration is conceptually interesting and may offer smoother generation trajectories.
The paper is mathematically detailed and ambitious.

**Weaknesses:**

1. There is no experiments, visual results, or empirical evidence. It reads more like a theoretical report than a balanced ICLR paper.
2. Writing is quite dense and formal, which makes it hard for general ML readers to follow the intuition or motivation.
3. Some sections (especially on circuit complexity) feel disconnected from the generative modeling goal.

**Questions:**

1. Can the authors include experimental or visual results to demonstrate the claimed improvements?
2. Consider providing a short intuitive overview before the more formal sections to guide readers through the technical content.

---

> ### Author Response · Authors · 2025-11-27
>
> Thank you very much for your valuable review. We will update in our next version.

---

### Official Review · Reviewer_yvy3 · 2025-11-01

**Soundness:** 3
**Presentation:** 2
**Contribution:** 3
**Rating:** 2
**Confidence:** 2

**Summary:**

This paper proposes the high-order pixel flow model HopeFlow, which is the first cascading flow model for end-to-end learning velocity field and acceleration field in pixel space. The model breaks through the limitations of the existing first-order supervision method. By integrating the characteristics of second-order dynamics, it effectively aligns the medium-range dependency and high-curvature region, and generates a smoother and more stable transmission trajectory.

**Strengths:**

1. Simplicity of end-to-end pixel-level generation: Abandoning the VAE encoder-decoder structure relied upon by latent variable diffusion models, HopeFlow directly performs multi-scale cascading flow matching in the original pixel space, avoiding information loss and additional computational overhead caused by latent variable space, while supporting end-to-end joint optimization, enhancing the model's interpretability and flexibility in tuning.

2. Innovation in high-order dynamics modeling: To address the trajectory instability issue caused by PixelFlow's modeling of only first-order velocity fields, HopeFlow introduces for the first time second-order acceleration fields into pixel-level flow matching. By jointly supervising velocity and acceleration, it provides explicit curvature guidance for high-curvature regions, significantly improving the smoothness and consistency of generated trajectories, and filling the gap in high-order dynamics modeling in existing flow matching methods.

**Weaknesses:**

1. Complementary comprehensive quantitative evaluation: add standard generation model metrics such as FID, IS, LPIPS, and compare with current models such as SD-XL, FLUX, etc., on public datasets like ImageNet, CIFAR-10, to clearly demonstrate HopeFlow's advantages in generation quality.

2. Lack of ablations and visualizations. I believe more figures and ablation experiments on public datasets can further improve this work.

**Questions:**

Please refer to the weaknesses.

---

> ### Author Response · Authors · 2025-11-27
>
> Thank you very much for your helpful comments. We will incorporate them in the next revision.

---

### Note · Authors · 2025-11-27

**Comment:**

I acknowledge and accept the venue's withdrawal policy on behalf of myself and my co-authors.

**Withdrawal Confirmation:**

I have read and agree with the venue's withdrawal policy on behalf of myself and my co-authors.